# Distributionally Robust Coreset Selection under Covariate Shift

**Tomonari Tanaka** *tanaka.tomonari.nagoyaml@gmail.com*
*Nagoya University*

**Hiroyuki Hanada**[*] *hanada.hiroyuki.i9@f.mail.nagoya-u.ac.jp*
*Nagoya University*
*RIKEN*

**Hanting Yang** *hanting.yang.nagoyaml@gmail.com*
*Nagoya University*

**Tatsuya Aoyama** *aoyama.tatsuya.nagoyaml@gmail.com*
*Nagoya University*

**Yu Inatsu** *inatsu.yu@nitech.ac.jp*
*Nagoya Institute of Technology*

**Satoshi Akahane** *akahane.satoshi.nagoyaml@gmail.com*
*Nagoya University*

**Yoshito Okura** *okura.yoshito.nagoyaml@gmail.com*
*Nagoya University*

**Noriaki Hashimoto** *noriaki.hashimoto.jv@riken.jp*
*RIKEN*

**Taro Murayama** *taro.murayama.j7z@jp.denso.com*
*DENSO CORPORATION*

**Hanju Lee** *lee.hanju.j8d@jp.denso.com*
*DENSO CORPORATION*

**Shinya Kojima** *shinya.kojima.j6s@jp.denso.com*
*DENSO CORPORATION*

**Ichiro Takeuchi**[†] *takeuchi.ichiro.n6@f.mail.nagoya-u.ac.jp*
*Nagoya University*
*RIKEN*

**Reviewed on OpenReview:** *https://openreview.net/forum?id=Eu7XMLJqsC*

## Abstract

Coreset selection, which involves selecting a small subset from an existing training dataset, is an approach to reducing training data size, and various approaches have been proposed for this method. In practical situations where these methods are employed, it is often the case that the data distributions differ between the development phase and the deployment phase, with the latter being unknown. Thus, it is challenging to select an effective subset

---

[*]Corresponding author.
[†]Corresponding author.

of training data that performs well across all deployment scenarios. We therefore propose Distributionally Robust Coreset Selection (DRCS), which theoretically derives an estimate of the upper bound for the worst-case test error, assuming that the future covariate distribution may deviate within a defined range from the training distribution. Furthermore, by selecting instances in a way that suppresses the estimate of the upper bound for the worst-case test error, DRCS achieves distributionally robust training instance selection. This study is primarily applicable to convex training computation, but we demonstrate that it can also be applied to deep learning under appropriate approximations. In this paper, we focus on covariate shift, a type of data distribution shift, and demonstrate the effectiveness of DRCS through experiments.

## 1 Introduction

Coreset selection is a technique designed to reduce the size of training data while maintaining the performance of predictive models (Guo et al., 2022). By carefully identifying a representative subset of the original dataset, this approach addresses critical challenges such as computational efficiency and memory limitations. Coresets are constructed to capture the most informative and diverse samples, ensuring that the essential characteristics of the underlying data distribution are preserved. This makes the method especially valuable in scenarios involving large-scale datasets or resource-constrained environments, where processing the entire dataset may be infeasible. Coreset selection finds broad applications in areas such as data summarization, accelerating model training, reducing annotation costs, and improving model interpretability. It is also practically applied to technologies that require retaining useful training instances, such as continual learning and active learning (Sener & Savarese, 2017; Toneva et al., 2019; Paul et al., 2021; Ducoffe & Precioso, 2018; Margatina et al., 2021). As machine learning advances into more complex domains, coreset selection will become increasingly critical for balancing efficiency and scalability.

This paper addresses the problem of coreset selection in scenarios where the future deployment environment of the model is uncertain. This challenge frequently arises in applications that require models to be tailored to specific, and often unpredictable, conditions or environments. For example, it includes tasks such as assessing investment risks under fluctuating economic conditions, predicting medical outcomes across diverse demographic groups, and estimating agricultural yields under varying climate scenarios. In such settings, the ability to create a representative and compact subset of the data that ensures reliable model performance is crucial. Traditional coreset selection does not make the assumption that future test distributions are uncertain, and it is unclear whether the model will work effectively under such conditions. The objective of this study is to develop a methodology for selecting a coreset from the original dataset while explicitly accounting for the need for worst-case robustness. This enables the model to perform effectively even under uncertain and variable future deployment conditions.

In this study, we focus on a distributionally robust setting under covariate shift conditions, which is a critical challenge in many real-world machine learning applications. Covariate shift occurs when the distribution of input features changes between the training and deployment (test) phases. Distributionally robust learning (Goh & Sim, 2010; Delage & Ye, 2010; Chen & Paschalidis, 2021) aims to tackle this problem by optimizing model performance under worst-case distributional shifts, enabling models to perform effectively across a wide range of potential data distributions and ensuring robustness to variability and uncertainties. Within this context, we address the specific problem of selecting a robust coreset, assuming that the future covariate distribution may deviate within a defined range from the distribution of the original dataset. To this end, we propose a novel method, termed the *Distributionally Robust Coreset Selection (DRCS)* method, which focuses on constructing a representative and robust subset of data tailored for these challenging conditions.

The basic idea of the DRCS method is to select a coreset that minimizes the worst-case test error under uncertain future test distributions, addressing the challenge of ensuring robustness in the face of potential distributional shifts. To achieve this, we derive a novel and theoretically grounded upper bound for the worst-case test error in the context of distributionally robust covariate shift. This upper bound serves as a critical foundation for guiding the coreset selection process. Building on this theoretical insight, we propose

an efficient algorithm designed to select a coreset that approximately minimizes this upper bound, ensuring that the resulting subset of data is both compact and robust to changes in future test distributions. Although the method is primarily developed for problems formulated within a specific class of convex optimization frameworks, it is versatile and can be extended to coreset selection for deep learning models by leveraging the neural tangent kernel (NTK) or fine-tuning scheme.

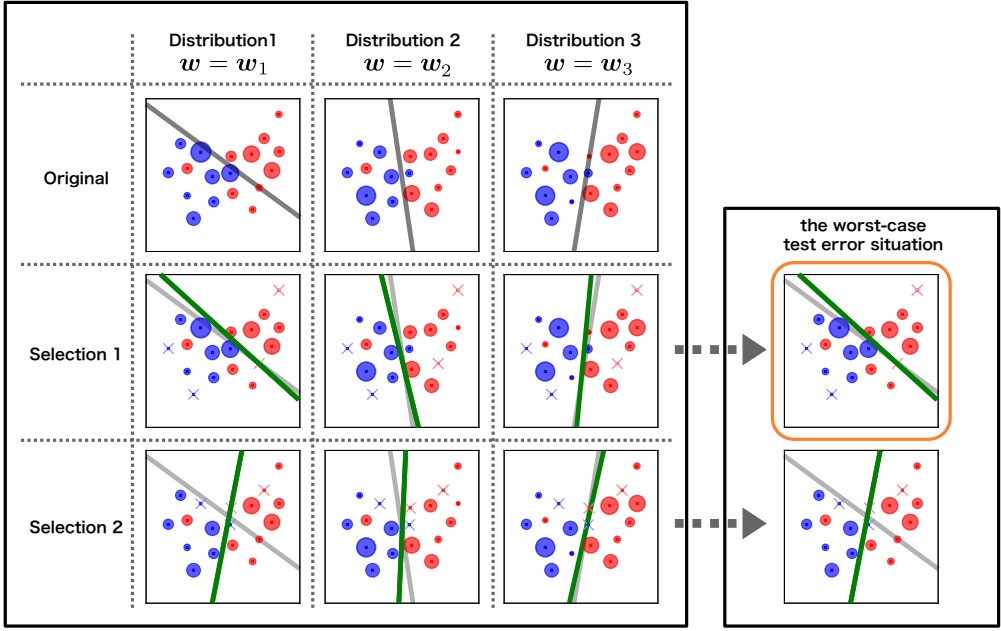

Figure 1: The concept of coreset selection in this study. In the left panel, each plot shows the distribution of the training data, where each column represents patterns of distribution changes, while each row represents patterns of instance selection. Let gray lines represent the learned results with specified weights and all instances, while green lines the retrained results with specified weights and selected instances. The goal is to obtain a selection pattern that can suppress the degradation of test error, even in the worst-case distribution for each selection pattern. In this figure, through the three distributions, Selection 1 can be considered a better choice than Selection 2. It is practically impossible since we need to explore such worst-case test error for all distributions and selection patterns.

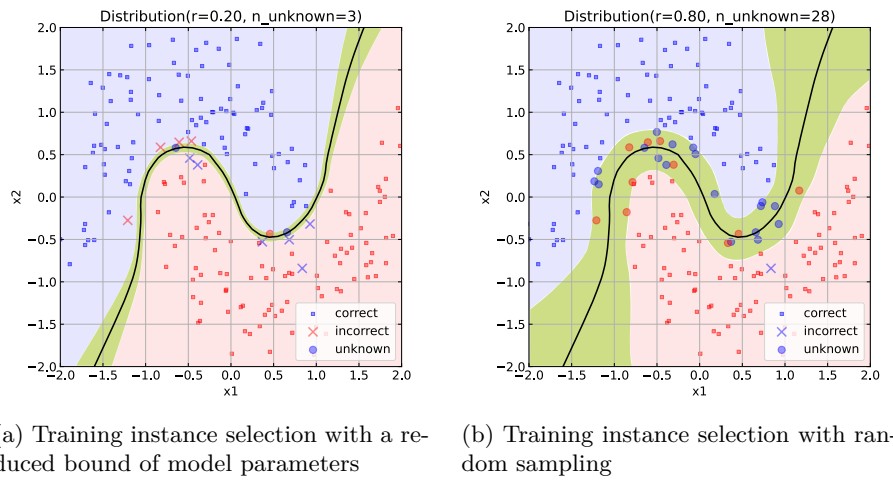

**[B] Our method (calculate a bound of model parameters)**

Figure 2: The concept of coreset selection in this study. These figures also show the distribution of the training data. The green area indicates the bound of model parameters obtained when retraining is performed, and we can calculate algorithmically the bound before retraining. We perform coreset selection to minimize the worst-case test error in the distribution, where the bound becomes the largest among all possible distributions.

(a) Training instance selection with a reduced bound of model parameters

(b) Training instance selection with random sampling

Figure 3: This figure illustrates an upper bound of the validation error in this study. Both figures show the distribution of the validation data. We calculate a bound of model parameters and use this to derive an upper bound of the validation error. In this figure, the blue and red areas represent that the validation data in these areas have a determined classification. On the other hand, the validation data in the green area does not have a determined classification. As a result, the upper bound of the validation error can be reached in cases where all instances in the green area are incorrectly classified. Since the bound of the model parameters depends on how to select instances such as (a) and (b), we perform coreset selection in a distribution where the upper bound of the validation error is minimized.

## 1.1 Contribution

In this study, we address the challenges discussed in Section 4 by introducing model parameter bounding techniques. Furthermore, we propose a distributionally robust coreset selection method that provides theoretical guarantees for model performance in binary classification problems. The contributions of this study are as:

- We consider the problem of coreset selection under covariate shift environments while taking distributional robustness into account, and propose a method to address this challenge.

- In the proposed method, we derive an upper bound of the validation error under the worst-case covariate shift and perform coreset selection to minimize this upper bound.

## 2 Problem Settings

In this section, we formulate a distributionally robust coreset selection (DRCS) problem in the context of uncertain covariate-shift settings.

### 2.1 Preliminaries and Notations

We consider a binary classification problem with the training dataset $\mathcal{D} := \{(\boldsymbol{x}_i, y_i)\}_{i \in [n]}$, where $n$ denotes the number of training instances, $\boldsymbol{x}_i \in \mathcal{X} \subseteq \mathbb{R}^d$ represents the $i$-th feature vector defined on the input domain $\mathcal{X}$, and $y_i \in \{\pm 1\}$ denotes the corresponding label for $i \in [n]$, with the notation $[n]$ indicating the set of natural numbers up to $n$. Similarly, let $\mathcal{D}' := \{(\boldsymbol{x}'_i, y'_i)\}_{i \in [n']}$ be the validation dataset of size $n'$, where each validation instance is assumed to follow the same distribution as the training instances. For binary classification problems, we consider a classifier parameterized by a set of parameters $\boldsymbol{\beta}$, defined as

$$f(\cdot; \boldsymbol{\beta}) : \mathbb{R}^d \ni \boldsymbol{x} \mapsto f(\boldsymbol{x}; \boldsymbol{\beta}) \in \mathbb{R}, \tag{1}$$

where the binary label for an input vector $\boldsymbol{x}$ is predicted as $-1$ if $f(\boldsymbol{x}; \boldsymbol{\beta}) < 0$, and $+1$ otherwise. Furthermore, we denote the loss function for binary classification (e.g., binary cross-entropy or hinge loss) as

$$\ell : \{\pm 1\} \times \mathbb{R} \ni (y, f(\boldsymbol{x}; \boldsymbol{\beta})) \mapsto \ell(y, f(\boldsymbol{x}; \boldsymbol{\beta})) \in \mathbb{R}_+, \tag{2}$$

where $\mathbb{R}_+$ indicates the set of nonnegative numbers [1].

In this study, we investigate distributionally robust learning under an uncertain covariate shift setting, where the input distributions for the training/validation datasets and the test datasets differ, with the discrepancy between these distributions known to lie within a specified range. In such a covariate-shift setting, it is well known that the difference in input distributions can be addressed through weighted learning. We denote the weight for the $i$-th training instance as $w_i > 0, i \in [n]$, and represent the $n$-dimensional vector of these weights as $\boldsymbol{w} \in [0, \infty)^n$. Typically, these weights are determined based on the ratio of the test input density to the training/validation input density (Shimodaira, 2000; Sugiyama et al., 2007). Therefore, a binary classification problem in a covariate-shift setting is generally formulated as a (regularized) weighted empirical risk minimization problem in the form of

$$\min_{\boldsymbol{\beta}} \frac{1}{\sum_{i \in [n]} w_i} \sum_{i \in [n]} w_i \ell(y_i, f(\boldsymbol{x}_i; \boldsymbol{\beta})) + \rho(\boldsymbol{\beta}), \tag{3}$$

---

[1]For example, in the case of the binary cross-entropy loss for a logistic regression model, it is expressed as

$$\ell(y, f(\boldsymbol{x}; \boldsymbol{\beta})) = \log(1 + e^{-yf(\boldsymbol{x}; \boldsymbol{\beta})}).$$

As another example, the hinge loss for a support vector machine (SVM) is expressed as

$$\ell(y, f(\boldsymbol{x}; \boldsymbol{\beta})) = \max\{0, 1 - yf(\boldsymbol{x}; \boldsymbol{\beta})\}.$$

where $\rho(\boldsymbol{\beta})$ denotes a regularization function.

In a conventional covariate-shift setting where the test input distribution is known, the weight vector $\boldsymbol{w}$ can be predetermined based on the density ratio, enabling the optimal model parameters to be obtained by directly solving the minimization problem in equation 3. On the other hand, in the distributionally robust setting, the density ratio and hence the weight vector $\boldsymbol{w}$ are unknown. In this study, we consider a distributionally robust learning scenario where the weight vector lies within a hypersphere of a certain radius $S$, centered around the uniform training data weights (i.e., $\boldsymbol{w} = \mathbf{1}_n := [1, \dots, 1]^\top$). This distributionally robust learning problem is formulated as

$$\min_{\boldsymbol{\beta}} \max_{\boldsymbol{w} \in \mathcal{W}} \frac{1}{\sum_{i \in [n]} w_i} \sum_{i \in [n]} w_i \ell(y_i, f(\boldsymbol{x}_i; \boldsymbol{\beta})) + \rho(\boldsymbol{\beta}), \tag{4}$$

where

$$\mathcal{W} := \{\boldsymbol{w} \in \mathbb{R}^n \mid \|\boldsymbol{w} - \mathbf{1}_n\|_2 \leq S\} \tag{5}$$

represents the hypersphere within which the weight vector can exist [2].

## 2.2 Applicable Class of Problems by the Proposed Method

Before presenting the proposed method, we define the class of classification problems to which the DRCS method applies. The proposed DRCS method is applicable when the classifier in equation 1, the loss function in equation 2, and the regularization function in equation 3 satisfy certain conditions. As a class of applicable classifiers, we focus on the basis function model in the form of

$$f(\boldsymbol{x}; \boldsymbol{\beta}) = \sum_{j \in [k]} \beta_j \phi_j(\boldsymbol{x}) = \boldsymbol{\beta}^\top \boldsymbol{\phi}(\boldsymbol{x}), \tag{6}$$

where $\phi_j : \mathbb{R}^d \ni \boldsymbol{x} \mapsto \phi_j(\boldsymbol{x}) \in \mathbb{R}, j \in [k]$ is the $j$-th basis function, $k$ is the number of basis functions, and $\boldsymbol{\phi}(\boldsymbol{x}) \in \mathbb{R}^k$ is the $k$-dimensional vector that gathers the $k$ basis functions. Furthermore, the loss function $\ell(y, f(\boldsymbol{x}; \boldsymbol{\beta}))$ is assumed to be convex with respect to its second argument, and the regularization function $\rho(\boldsymbol{\beta})$ must also be convex with respect to $\boldsymbol{\beta}$.

While these conditions may seem restrictive, kernelization is achievable by considering the dual form of the basis function model in equation 6, enabling extensions to popular nonlinear classifiers such as nonlinear logistic regression and kernel support vector machines. When applying the proposed method to deep learning models, tools such as Neural Tangent Kernel (NTK) (Novak et al., 2020) and Neural Network Gaussian Processes (NNGP) (Lee et al., 2017) can be utilized. These tools bridge traditional kernel methods with deep learning, offering both theoretical insights and practical tools for tackling high-dimensional and complex learning problems. Furthermore, in practical DRCS problems, since the training and test data are typically assumed to share a certain degree of similarity, a fine-tuning approach that updates part of the model parameters during testing is beneficial. The proposed method is applicable in such scenarios and serves as a valuable tool for coreset selection in deep learning models as well.

## 2.3 Distributionally Robust Coreset Selection (DRCS) Problems

To formulate the coreset selection problem, let us introduce an $n$-dimensional binary vector $\boldsymbol{v} \in \{0, 1\}^n$, where $v_i = 1$ indicates that the $i$-th training instance is included in the coreset, while $v_i = 0$ indicates that the $i$-th training instance is excluded. Hereafter, we refer to $\boldsymbol{v}$ as the *coreset vector*. Let us write the training error of our DRCS problem as

$$P_{\boldsymbol{v},\boldsymbol{w}}(\boldsymbol{\beta}) := \frac{1}{\sum_{i \in [n]} v_i w_i} \sum_{i \in [n]} v_i w_i \ell(y_i, f(\boldsymbol{x}_i; \boldsymbol{\beta})) + \rho(\boldsymbol{\beta}). \tag{7}$$

---

[2]We assume $S \leq 1$, focusing on scenarios where the differences between the training and test input distributions are not substantial.

Given a coreset vector $\boldsymbol{v} \in \{0,1\}^n$ and a weight vector $\boldsymbol{w} \in \mathcal{W}$, the classifier's optimal model parameter vector is written as

$$\boldsymbol{\beta}^*(\boldsymbol{v}, \boldsymbol{w}) := \arg\min_{\boldsymbol{\beta}} P_{\boldsymbol{v}, \boldsymbol{w}}(\boldsymbol{\beta}). \tag{8}$$

Given a weight vector $\boldsymbol{w}' \in \mathbb{R}^{n'}$ for validation dataset, the weighted validation error for the optimal model parameter vector in equation 8 can be defined as

$$\mathrm{VaEr}(\boldsymbol{v}, \boldsymbol{w}, \boldsymbol{w}') := \frac{1}{\sum_{i \in [n']} w'_i} \sum_{i \in [n']} w'_i I\left\{ y'_i \neq \mathrm{sgn}(f(\boldsymbol{x}'_i; \boldsymbol{\beta}^*(\boldsymbol{v}, \boldsymbol{w}))) \right\}, \tag{9}$$

where

$$\mathcal{W}' := \{ \boldsymbol{w}' \in \mathbb{R}^{n'} \mid \|\boldsymbol{w}' - \mathbf{1}_{n'}\|_2 \leq Q \} \tag{10}$$

represents the hypersphere of a certain radius $Q$ within which the weight vector can exist. Here, $I\{\cdot\}$ is the indicator function that returns 1 if the argument is true and 0 otherwise, and $\mathrm{sgn} : \mathbb{R} \to \{\pm 1\}$ is the sign function that returns the sign of the given scalar input. If the weight vector $\boldsymbol{w}'$ for the validation dataset is appropriately determined based on the density ratio of the input distributions of the validation and the test datasets, the weighted validation error VaEr in equation 9 can be used as a test error estimator.

In a distributionally robust setting where the weight vector is uncertain, minimizing the worst-case test error is necessary. As an estimator of the worst-case test error, it is reasonable to use the worst-case weighted validation error (WrVaEr) in equation 9, expressed as

$$\mathrm{WrVaEr}(\boldsymbol{v}) = \max_{\boldsymbol{w}' \in \mathcal{W}'} \max_{\boldsymbol{w} \in \mathcal{W}} \mathrm{VaEr}(\boldsymbol{v}, \boldsymbol{w}, \boldsymbol{w}'). \tag{11}$$

Based on the above discussion, the goal of the DRCS problem is formulated as the problem of finding the coreset vector $\boldsymbol{v} \in \{0,1\}^n$ that minimizes the worst-case weighted validation error in equation 11. Let $m < n$ be the size of the coreset, i.e., the number of remaining training instances after coreset selection. Then, the DRCS problem we consider is formulated as

$$\boldsymbol{v}^* = \arg\min_{\boldsymbol{v}} \mathrm{WrVaEr}(\boldsymbol{v}) \text{ subject to } \|\boldsymbol{v}\|_1 = m. \tag{12}$$

By summarizing equations 8, 9, 11 and 12, minimizing the worst-case weighted validation error (WrVaEr) can be expressed as:

$$\min_{\boldsymbol{v}} \max_{\boldsymbol{w}'} \max_{\boldsymbol{w}} \min_{\boldsymbol{\beta}} \mathrm{VaEr}(\boldsymbol{v}, \boldsymbol{w}, \boldsymbol{w}', \boldsymbol{\beta}). \tag{13}$$

Unfortunately, the problem in equation 12 is highly challenging. First, it involves complex optimization over four types of vectors: $\boldsymbol{v}$, $\boldsymbol{w}$, $\boldsymbol{w}'$, and $\boldsymbol{\beta}$. This optimization is technically complex as it requires solving nested optimization problems across four hierarchical levels, involving a large number of variables and significantly increasing computational complexity, along with the difficulty of ensuring convergence and global optimality. Moreover, the fourth-level optimization for $\boldsymbol{v} \in \{0,1\}^n$ is a combinatorial optimization problem, which becomes infeasible to solve optimally when $n$ is large. To address these technical challenges, our approach involves the following two steps:

(i) Deriving an upper bound for the worst-case weighted validation error in equation 11.

(ii) Greedily identifying a coreset that minimizes this upper bound of worst-case weighted validation error.

In the next section, we describe methods for (i) and (ii) in Section 3.1 and Section 3.2, respectively.

## 2.4 A Working Example: $L_2$-regularized Logistic Regression

As a working example of such a classification problem, let us consider $L_2$-regularized logistic regression of basis function models, i.e., we consider a basis function model in equation 6, binary cross-entropy as the loss function $\ell(y, f(\boldsymbol{x}; \boldsymbol{\beta}))$ and $L_2$ regularization function as the regularization function $\rho(\boldsymbol{\beta})$. Considering the coreset vector $\boldsymbol{v}$, the optimization problem for $L_2$-regularized logistic regression is defined as

$$\boldsymbol{\beta}^*_{\boldsymbol{v}, \boldsymbol{w}} = \underset{\boldsymbol{\beta} \in \mathbb{R}^k}{\arg\min}\, P_{\boldsymbol{v}, \boldsymbol{w}}(\boldsymbol{\beta}), \tag{14}$$

where

$$P_{\boldsymbol{v}, \boldsymbol{w}}(\boldsymbol{\beta}) = \frac{1}{\sum_{i \in [n]} v_i w_i} \sum_{i \in [n]} v_i w_i \log(1 + e^{-y_i f(\boldsymbol{x}_i; \boldsymbol{\beta})}) + \frac{\lambda}{2}\|\boldsymbol{\beta}\|_2^2, \tag{15}$$

which is referred to as the *primal problem* and $P_{\boldsymbol{v}, \boldsymbol{w}}(\boldsymbol{\beta})$ is called *primal objective function*.

# 3 Proposed DRCS Method

In this section, we present the proposed method to solve DRCS problem in equation 12. As discussed in the previous section, since it is difficult to directly minimize the validation error (equation 11), we show how to derive and minimize its upper bound. Also, since it still includes difficult problems (a combinatorial optimization and an alternative optimization), we show specific algorithm to solve it using a greedy optimization.

## 3.1 Upper Bound for the Worst-case Test Error

In this subsection, we present our main result.

### 3.1.1 Main Theorem

An upper bound of the worst-case weighted validation error, which can serve as the worst-case test error estimator, WrVaEr($\boldsymbol{v}$) in equation 11, is presented in the following theorem. Our main result can be formulated as equation 18.

**Definition 3.1.** *For a convex function $f : \mathbb{R}^n \to \mathbb{R}$, its* convex conjugate *$f^* : \mathbb{R}^n \to \mathbb{R} \cup \{+\infty\}$ is defined as $f^*(\boldsymbol{\beta}) = \sup_{\boldsymbol{\beta}' \in \mathbb{R}^n}(\boldsymbol{\beta}^\top \boldsymbol{\beta}' - f(\boldsymbol{\beta}'))$.*

**Definition 3.2.** *A function $f : \mathbb{R}^n \to \mathbb{R}$ is called $\mu$-strongly convex if $f(\boldsymbol{\beta}) - \frac{\mu}{2}\|\boldsymbol{\beta}\|_2^2$ is convex.*

**Definition 3.3.** *For $\arg\min_{\boldsymbol{\beta}} P_{\boldsymbol{v}, \boldsymbol{w}}(\boldsymbol{\beta})$ in equation 7, we define $f(\boldsymbol{x}; \boldsymbol{\beta}) = \boldsymbol{\beta}^\top \boldsymbol{\phi}(\boldsymbol{x})$. Then, its* dual problem *is defined as*

$$D_{\boldsymbol{v}, \boldsymbol{w}}(\boldsymbol{\alpha}) = -\frac{1}{\sum_{i \in [n]} v_i w_i} \sum_{i \in [n]} v_i w_i \ell^*(-\alpha_i) - \rho^*\left(\frac{1}{\sum_{i \in [n]} v_i w_i}(\mathrm{diag}(\boldsymbol{v} \otimes \boldsymbol{w} \otimes \boldsymbol{y})\Phi)^\top \boldsymbol{\alpha}\right). \tag{16}$$

*where $\Phi := \begin{bmatrix} \boldsymbol{\phi}(\boldsymbol{x}_1) & \boldsymbol{\phi}(\boldsymbol{x}_2) \ldots \boldsymbol{\phi}(\boldsymbol{x}_n) \end{bmatrix}^\top \in \mathbb{R}^{n \times k}$, and $\otimes$ is the element-wise product. Here, especially, if $\rho(\boldsymbol{\beta}) := \frac{\lambda}{2}\|\boldsymbol{\beta}\|_2^2$, it is calculated as*

$$D_{\boldsymbol{v}, \boldsymbol{w}}(\boldsymbol{\alpha}) = -\frac{1}{\sum_{i \in [n]} v_i w_i} \sum_{i \in [n]} v_i w_i \ell^*(-\alpha_i) - \frac{1}{2\lambda\left(\sum_{i \in [n]} v_i w_i\right)^2}\|(\mathrm{diag}(\boldsymbol{v} \otimes \boldsymbol{w} \otimes \boldsymbol{y})\Phi)^\top \boldsymbol{\alpha}\|_2^2. \tag{17}$$

This derivation is presented in Appendix B.1.

Then we state the main theorem as follows.

**Theorem 3.4.** *Assume that $\rho$ in the primal objective function $P_{\mathbf{1}_n,\mathbf{1}_n}$ is $\mu$-strongly convex with respect to $\boldsymbol{\beta}$. Let us denote the optimal primal and dual solutions for the entire training set (i.e., $v_i = 1 \ \forall i \in [n]$) with uniform weights (i.e., $w_i = 1 \ \forall i \in [n]$) as*

$$\boldsymbol{\beta}^*_{\mathbf{1}_n,\mathbf{1}_n} = \arg\min_{\boldsymbol{\beta}\in\mathbb{R}^k} P_{\mathbf{1}_n,\mathbf{1}_n}(\boldsymbol{\beta}) \quad and \quad \boldsymbol{\alpha}^*_{\mathbf{1}_n,\mathbf{1}_n} = \arg\max_{\boldsymbol{\alpha}\in\mathbb{R}^n} D_{\mathbf{1}_n,\mathbf{1}_n}(\boldsymbol{\alpha}),$$

*respectively. Then, an upper bound of the worst-case weighted validation error is written as*

$$\mathrm{WrVaEr}(\boldsymbol{v}) \leq \mathrm{WrVaEr}^{\mathrm{UB}}(\boldsymbol{v}) = 1 - \left( \mathbf{1}_{n'}^\top \boldsymbol{\zeta}(\boldsymbol{v}) - Q\sqrt{\|\boldsymbol{\zeta}(\boldsymbol{v})\|_2^2 - \frac{(\mathbf{1}_{n'}^\top\boldsymbol{\zeta}(\boldsymbol{v}))^2}{n'}} \right) \frac{1}{n'}, \tag{18}$$

*where,*

$$\boldsymbol{\zeta}(\boldsymbol{v}) \in \{0,1\}^{n'}; \quad \zeta_i(\boldsymbol{v}) = I\left\{ y_i' \boldsymbol{\beta}^{*\top}_{\mathbf{1}_n,\mathbf{1}_n} \boldsymbol{\phi}(\boldsymbol{x}_i') - \|\boldsymbol{\phi}(\boldsymbol{x}_i')\|_2 \sqrt{\frac{2}{\lambda} \max_{\boldsymbol{w}\in\mathcal{W}} \mathrm{DG}(\boldsymbol{v},\boldsymbol{w})} > 0 \right\}, \tag{19}$$

$$\mathrm{DG}(\boldsymbol{v},\boldsymbol{w}) := P_{\boldsymbol{v},\boldsymbol{w}}(\boldsymbol{\beta}^*_{\mathbf{1}_n,\mathbf{1}_n}) - D_{\boldsymbol{v},\boldsymbol{w}}(\boldsymbol{\alpha}^*_{\mathbf{1}_n,\mathbf{1}_n}). \tag{20}$$

*Here, the quantity* $\mathrm{DG}$ *is called the* duality gap *and it plays a main role of the upper bound of the validation error.*

*Especially, if we use L2-regularization $\rho(\beta) := \frac{\lambda}{2}\|\boldsymbol{\beta}\|_2^2$, $\mathrm{DG}(\boldsymbol{v},\boldsymbol{w})$, the maximization $\max_{\boldsymbol{w}\in\mathcal{W}} \mathrm{DG}(\boldsymbol{v},\boldsymbol{w})$ can be algorithmically computed.*

We sketch the proof of Theorem 3.4 in the next section. The complete proof is given in Appendix C.1.

### 3.1.2 Proof Sketch

The proof sketch of Theorem 3.4 is primarily divided into the following steps.

1. First, as a premise, consider a model where $P_{\boldsymbol{w}}$ is $\mu$-strongly convex.

2. Second, under the point 1 above, derive the bound of model parameters. The model parameter vector $\boldsymbol{\beta}^*_{\boldsymbol{v},\boldsymbol{w}}$, trained using the coreset vector $\boldsymbol{v}$ and weight vector $\boldsymbol{w}$, is guaranteed to converge within the range $\mathcal{B}^*_{\boldsymbol{v},\boldsymbol{w}}$, which is represented by an L2-norm hypersphere (Hanada et al., 2023). A hypersphere of the radius is given by $R = \sqrt{\frac{2}{\lambda}\mathrm{DG}(\boldsymbol{v},\boldsymbol{w})}$, which is calculated in equation 20.

3. Third, calculate a bound of the weighted validation error. Using the hypersphere range $\mathcal{B}^*_{\boldsymbol{v},\boldsymbol{w}}$, computed for a specific coreset vector $\boldsymbol{v}$ and weight vector $\boldsymbol{w}$, an upper bound of the weighted validation error can be analytically calculated. This upper bound is determined by DG. The larger DG, the greater the upper bound becomes.

4. Fourth, maximize DG with respect to the weight vector $\boldsymbol{w}$ for training dataset in equation 21. We can obtain an upper bound of the worst-case weighted validation error. We can easily confirm that $\mathrm{DG}(\boldsymbol{v},\boldsymbol{w})$ is a convex function with respect to $\boldsymbol{w}$, and therefore its maximization is difficult in general. However, if we use L2-regularization, it becomes a convex quadratic function and its maximization can be algorithmically computed by solving an eigenvalue problem. In fact,

$$\max_{\boldsymbol{w}\in\mathcal{W}} \mathrm{DG}(\boldsymbol{v},\boldsymbol{w}) = \max_{\boldsymbol{w}\in\mathcal{W}} (\boldsymbol{v}\otimes\boldsymbol{w})^\top A (\boldsymbol{v}\otimes\boldsymbol{w}) + \boldsymbol{b}^\top (\boldsymbol{v}\otimes\boldsymbol{w}) + c, \tag{21}$$

   where the matrix $A$ of the quadratic term, the vector $\boldsymbol{b}$ of the linear term, and the constant term $c$ are respectively given as

$$A = \frac{1}{2\lambda} \mathrm{diag}(\boldsymbol{\alpha}^*_{\mathbf{1}_n,\mathbf{1}_n}\otimes\boldsymbol{y})^\top K \mathrm{diag}(\boldsymbol{\alpha}^*_{\mathbf{1}_n,\mathbf{1}_n}\otimes\boldsymbol{y}),$$

$$\boldsymbol{b} = \left[ \ell(y_i, f(\boldsymbol{x}_i; \boldsymbol{\beta}^*_{\mathbf{1}_n,\mathbf{1}_n})) - \alpha^*_{\mathbf{1}_n,\mathbf{1}_n,i} \right]_{i\in[n]},$$

$$c = \frac{1}{2\lambda}(\mathbf{1}_n\otimes\boldsymbol{\alpha}^*_{\mathbf{1}_n,\mathbf{1}_n}\otimes\boldsymbol{y})^\top K(\mathbf{1}_n\otimes\boldsymbol{\alpha}^*_{\mathbf{1}_n,\mathbf{1}_n}\otimes\boldsymbol{y}).$$

   Here, $K \in \mathbb{R}^{n\times n}$ is a kernel matrix, where is defined as $K_{i,j} = \boldsymbol{\phi}(\boldsymbol{x}_i)^\top \boldsymbol{\phi}(\boldsymbol{x}_j)$.

5. Finally, maximize an upper bound of weighted validation error with respect to the weight vector $\boldsymbol{w}'$ for the validation dataset. We can derive an upper bound of the worst-case weighted validation error, leading to equation 19 and therefore equation 18.

### 3.2 Greedy Coreset Selection Based on the Upper Bound

The basic idea of the proposed DRCS method is to select the coreset vector $\boldsymbol{v} \in \{0,1\}^n$ that minimizes the upper bound of the worst-case weighted validation error represented by equation 18. Since this is a combinatorial optimization problem, finding the global optimal solution within a realistic timeframe is challenging; thus, we adopt greedy approaches to obtain approximate solutions.

A naive greedy approach, referred to as `greedy approach 1`, repeats the followings: we remove one training instance that minimizes equation 18, and update $\boldsymbol{w}$ to maximize $\mathrm{DG}(\boldsymbol{v}, \boldsymbol{w})$ in equation 20 (Algorithm 1). Although `greedy approach 1` is sufficient for small datasets, the computational cost becomes prohibitively high for larger datasets. The most significant computational cost in calculating the bound in equation 18 lies in the eigendecomposition of $A$ in equation 21, required to determine $\max_{\boldsymbol{w} \in \mathcal{W}} \mathrm{DG}(\boldsymbol{v}, \boldsymbol{w})$, which needs $O(n^3)$ time with a naive computation, or $O(n^2)$ if it is approximated. To circumvent this cost, another approach, referred to as `greedy approach 2`, does not update $\boldsymbol{w}$ whenever an instance is removed, but instead fixes $\boldsymbol{w}$ optimized with initial $\boldsymbol{v}$ (i.e., $\boldsymbol{v} = \mathbf{1}_n$) (Algorithm 2). Then, it recalculates the values of $\min_{\boldsymbol{v}} \mathrm{WrVaEr}^{\mathrm{UB}}(\boldsymbol{v})$ after the removal of each instance to dynamically update the selection process. Moreover, as a much simpler approach, referred to as `greedy approach 3`, determines the instances to be removed based solely on the initial values of $\min_{\boldsymbol{v}} \mathrm{WrVaEr}^{\mathrm{UB}}(\boldsymbol{v})$ without recalculating them after each removal (Algorithm 3). These greedy approaches are heuristics and do not guarantee optimality. However, in Section 5, we demonstrate that these approaches facilitate the selection of a coreset, effectively mitigating the increase in worst-case test error.

The time complexities of these algorithms are as follows (if eigendecomposition is not approximated), where $n^{\mathrm{del}}$ is the number of deletions: $O(n^4 n^{\mathrm{del}})$ for greedy approach 1, $O(n^3 n^{\mathrm{del}})$ for greedy approach 2, and $O(n^3 + n^{\mathrm{del}})$ for greedy approach 3. The details of these algorithm is given in Appendix C.4. In this Appendix, we provide the pseudocode of them.

## 4 Related Works and Limitations

### 4.1 Related Works

**Coreset Selection.** Coreset selection is a technique for selecting important data samples in training to enhance data efficiency, reduce the computational cost, and maintain or improve model accuracy. Currently, several approaches exist, each differing in how they evaluate the importance of data. Representative methods are outlined below. **Geometry-Based Methods:** Geometry-Based Methods utilize the data distribution in the feature space to improve learning efficiency by reducing redundant samples. Examples include k-Center-Greedy(Sener & Savarese, 2017), which minimizes the maximum distance between samples, and Herding(Welling, 2009), which iteratively selects samples based on the distance between the coreset center and the original dataset center. **Uncertainty-Based Methods:** Uncertainty-Based Methods prioritize selecting samples for which the model has the least confidence. Methods such as Least Confidence, Entropy, and Margin select high-uncertainty samples based on these metrics(Coleman et al., 2019). **Error/Loss-Based Methods:** Error/Loss-Based Methods select important samples based on loss function values or gradient information. Examples include GraNd(Paul et al., 2021) and EL2N(Paul et al., 2021), which are based on the magnitude of the loss. **Sensitivity-Based Methods:** Sensitivity-Based Methods are similar to Error/Loss-Based Methods, however, for each sample, its importance value is defined by the maximum of the relative loss (ratio of the loss of the sample to the sum of the losses of all samples) among all possible model parameters. It can assure the upper bound of the model parameter shifts by the coreset selection in more strict manner (called the $\varepsilon$-coreset) (Munteanu et al., 2018; Tukan et al., 2020; 2021; Tolochinksy et al., 2022; Alishahi & Phillips, 2024). **Decision Boundary-Based Methods:** Decision Boundary-Based Methods focus on selecting samples near decision boundaries that are difficult to classify. Examples include Adversarial DeepFool(Ducoffe & Precioso, 2018) and Contrastive Active Learning (CAL)(Margatina et al.,

2021). **Gradient Matching-Based Methods:** Gradient Matching-Based Methods aim to approximate the gradient of the entire dataset with a small number of samples. CRAIG(Mirzasoleiman et al., 2020) and GradMatch(Killamsetty et al., 2021a) utilize this approach by leveraging gradient information. **Bilevel Optimization-Based Methods:** Bilevel Optimization-Based Methods formulate coreset selection as a bilevel optimization problem. Retrieve(Killamsetty et al., 2021c) and Glister(Killamsetty et al., 2021b) have been applied to continual learning and active learning. **Submodularity Optimization-Based Methods:** Optimization methods based on Submodular functions(Iyer & Bilmes, 2013) enable combinatorial optimization by introducing submodular functions to avoid the combinatorial explosion, allowing for the optimization of diverse sets of samples. Examples of submodular functions include Graph Cut (GC), Facility Location (FL), and Log-Determinant (LD)(Iyer et al., 2021). Other approaches can be found, for example, in Guo et al. (2022).

**Distributionally Robust.** DR has been studied in various machine learning problems to enhance model robustness against variations in data distribution. The DR learning problem is generally formulated as a worst-case optimization problem to account for potential distributional shifts. Consequently, techniques that integrate DR learning and optimization have been explored in both fields. The proposed method builds upon such DR techniques, emphasizing effective training sample selection even when the future test distribution is unknown. It incorporates DR considerations during sample selection rather than during training computation. While the primary goal of the proposed method is to reduce computational resources through sample removal, this process also has practical implications in other scenarios. For example, in continual learning (e.g., see Wang et al. (2022)), managing data by selectively retaining or discarding samples is crucial, especially when anticipating shifts in future data distributions. Improper deletion of important data may result in catastrophic forgetting(Kirkpatrick et al., 2017), where the model loses previously acquired knowledge after learning on new data. Our proposed coreset selection method explicitly addresses this DR setting, and, to the best of our knowledge, no existing studies explore coreset selection under such a framework. Moreover, many existing coreset selection methods are heuristic in nature and lack robust theoretical guarantees.

## 4.2 Limitations

The proposed DRCS method is developed for binary classification problems and is applicable to models (such as SVM and logistic regression) where the primal objective function (loss function + regularization term) possesses strong convexity. Therefore, it cannot be directly applied to deep learning models. However, by appropriately selecting a regularization function, the method can also be extended to kernel methods. Consequently, we can well approximate the deep learning by utilizing the recently popular Neural Tangent Kernel(NTK)(Novak et al., 2020).

The proposed DRCS method utilizes a bound of the model parameter, and a similar approach has been studied in instance selection by Hanada et al. (2023). The method proposed by Hanada et al. (2023) (DRSSS) is effective for sample-sparse models due to its characteristics, but it is limited to models that are both strongly convex and instance-sparse, and can remove only samples that do not change the training results at all.

In contrast, the proposed method is applicable to any model with strong convexity, making it a more versatile approach.

## 5 Numerical Experiment

### 5.1 Experimental Settings

In this section, we numerically evaluate the proposed DRCS through experiments. For this experiment, $\boldsymbol{\beta}^*_{\mathbf{1}_n, \mathbf{1}_n}, \boldsymbol{\alpha}^*_{\mathbf{1}_n, \mathbf{1}_n}$ is learned by solving equation 7 and 16. We perform cross-validation with a training-to-test data ratio of 4:1 in the experiments. Next, we define the training weight range as $\mathcal{W} := \{\boldsymbol{w} \mid \|\boldsymbol{w} - \mathbf{1}_n\|_2 \leq S\}$ and set $S$ as follows. The datasets are primarily designed for binary classification, and for multi-class datasets, two classes are extracted and used. We set $S$ with a parameter $a$ as $S = \sqrt{n^+}|a-1|$, where $n^+$ is the number

of positive instances in the training dataset. We defined as above since it is equivalent to the case when the class balance is changed; if the weights of all positive instances ($i \mid y_i = +1$) change from 1 to $a$ while the weights for negative instances ($i \mid y_i = -1$) remain at 1, $S$ is calculated as above. $a = 1$ means that no distribution change occurs. Similarly, we define the validation weight range as $\mathcal{W}' := \{\boldsymbol{w}' \mid \|\boldsymbol{w}' - \mathbf{1}_n\|_2 \leq Q\}$ and set $Q$ in the same way as above. In the learning setup of this paper, we adopt the empirical risk minimization approach as shown in equation 7, and thus the regularization parameter $\lambda$ is determined based on the number of instances. Specifically, we first present results obtained by retraining with $\lambda$ that is determined by cross-validation. Results for other values of $\lambda$ will be discussed later in Section 5.4. Details of implementations are presented in Appendix D.1, and details of data setups and hyperparameters are presented in Appendix D.2.

## 5.2 Coreset Selection for Tabular Data

In this section, we present the experimental results for tabular datasets. The experiments in this section were conducted using a logistic regression model (logistic loss + L2 regularization), with Radial Basis Function (RBF) kernels [3] (Schölkopf et al., 2001) applied to the datasets listed in Table 1. In this experiment for tabular data, we adopt Algorithm 1 for calculation of the upper bound in equation 18.

**Baselines.** The baseline methods for comparing coreset selection are as follows: geometry based methods: (a)Herding (Welling, 2009), (b)k-Center-Greedy (Sener & Savarese, 2017), uncertainty-based methods: (c) Margin(:a method that selects instances closest to the decision boundary (margin).) Other method: (d)Random sampling. For table datasets, methods that do not rely on deep learning are chosen.

The results are shown in Figure 4 and 5. The horizontal axis represents the size of the selected dataset, where moving to the right indicates a smaller selected dataset. The vertical axis represents the weighted validation accuracy ($1 - \text{VaEr}$) minimized with respect to $\boldsymbol{w}'$ by using equation 43. The retraining settings of all methods are controlled to be the same. In Figure 4, the proposed method demonstrates the higher weighted validation accuracy compared to other methods. We also found that, depending on setups, existing methods (that do not consider DR) may end in worse performance than random sample removal. This implies that, to achieve good coreset selection under DR, the consideration of DR is essential. All results for other datasets, other $\lambda$ and support vector machine model are presented in Appendix D.3 and D.4.

## 5.3 Coreset Selection for Image Data

This section discusses coreset selection for image datasets. As discussed in Section 2.2, although the proposed method is not specifically designed for deep learning, it can be applied to image data by leveraging Neural Tangent Kernels (NTK)(Novak et al., 2020) or using the layers preceding the final layer of a deep learning model as a feature extractor. In this experiment, considering future extensibility, we apply the proposed method to features extracted from images using a feature extractor. The experimental results using NTK are included in the Appendix D.6. In this paper, we evaluate the DRCS method using the CIFAR10 dataset (Krizhevsky, 2009). CIFAR10 consists of 50,000 training instances and 10,000 test instances, divided into 10 categories. Since the proposed method is designed for binary classification, we extract a subset of the CIFAR10 dataset. Specifically, we create a binary classification dataset consisting of 40,000 training instances categorized as vehicles (airplane, automobile, ship, truck) and animals (cat, deer, dog, horse). The test dataset is similarly divided into two classes, resulting in a total of 8,000 test instances. The generalization performance of the datasets selected by the coreset selection methods is evaluated using the widely used deep neural network ResNet50 (He et al., 2016). In this experiment, all hyperparameters and experimental settings are kept consistent before and after instance selection and model retraining. Specifically, for all experiments, we use a batch size of 128, a learning rate of 0.01, weight decay of 0.001, and train the model using the Adam optimizer for 100 epochs. In the proposed method, the selected instances are transformed using a fixed feature extractor, and classification is performed using the DRCS algorithm. The feature extractor is trained under the same settings as described in the earlier experimental setup. In this experiment for image data, we adopt Algorithm 2 for calculation of the bound in equation 18.

---

[3]Hyperparameters of RBF kernel are determined with heuristics; see Appendix D.2

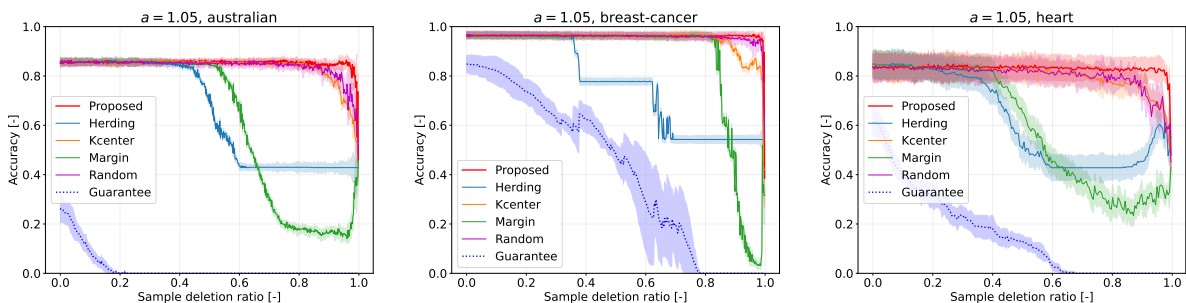

Figure 4: We compare our proposed method with several instance selection baselines with respect to the weighted validation accuracy $(1 - \text{VaEr})$. Our method exhibits superior performance.

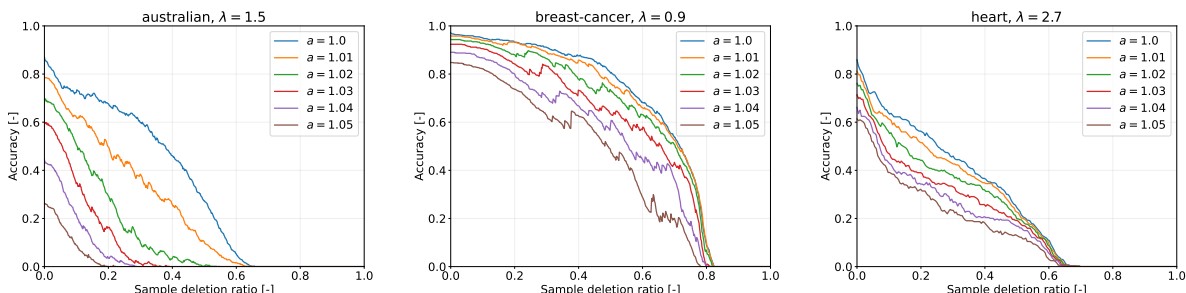

Figure 5: We show the lower bound of the worst-case weighted validation accuracy $(1 - \text{WrVaEr}^{\text{UB}})$. This graph indicates that the lower bound of test accuracy is theoretically guaranteed under covariate shift with an unknown test distribution.

**Baselines.** The baseline methods for comparing coreset selection are as follows: uncertainty-based method:(a)Least Confidence (Coleman et al., 2019), error/loss-based methods: (b)GraNd (Paul et al., 2021), (c)DeepFool (Ducoffe & Precioso, 2018), Gradient matching-based method: (d)GradMatch (Killamsetty et al., 2021a), Bilevel optimization-based method: (e)Glister (Killamsetty et al., 2021b), Submodularity-based method: (f)Log-Determinant (Iyer et al., 2021). For image datasets, comparative experiments were conducted, including methods designed for deep learning. In these experiments, the baseline methods were implemented using Guo et al. (2022).

The results are shown in Figure 6. The method for visualization and calculation is the same as that used in Section 5.2. In this figure, the proposed method demonstrates the higher weighted validation accuracy compared to other methods.

## 5.4 Discussion on Regularization Parameters

In the previous section, we presented results using a specific preset value for $\lambda$ based on a predefined criterion. Ideally, $\lambda$ should be selected to optimize model performance. Here, we discuss the relationship between the choice of $\lambda$, the level of theoretical guarantees, and the resulting model performance. Here, we present the results for table data.

Figure 7 illustrates the number of removed instances that can be theoretically guaranteed for specific accuracy levels across different values of $\lambda$. First, let us compare the columns in the top row of Figure 7. In the first and fourth columns, it can be observed that the performance of the proposed method deteriorates in the later stages. When strong regularization is applied, the model's performance remains poor even with the entire training set, which diminishes the effectiveness of the proposed method. On the other hand, when the regularization is small, the bound of the model parameters becomes broader, making instance selection less effective.

Table 1: Tabular datasets for numerical experiments. All are binary classification datasets from LIBSVM dataset (Chang & Lin, 2011).

| Name | $n$ | $n^+$ | $d$ |
|---|---|---|---|
| splice | 1000 | 517 | 61 |
| australian | 690 | 307 | 15 |
| breast-cancer | 683 | 239 | 11 |
| heart | 270 | 120 | 14 |
| ionosphere | 351 | 225 | 35 |

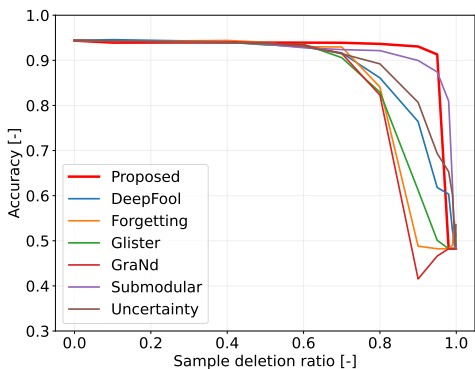

Figure 6: We compare our proposed method with several instance selection baselines with respect to the weighted validation accuracy $(1 - \mathrm{VaEr})$. Our method exhibits superior performance generally.

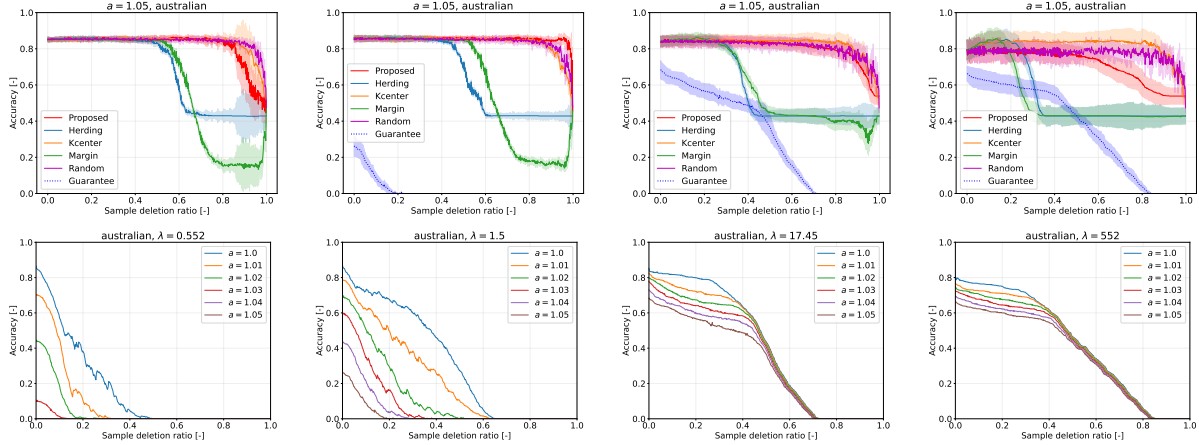

Figure 7: The results represent the model performance across varying values of lambda. The top row corresponds to the weighted validation accuracy $(1 - \mathrm{VaEr})$, and the bottom row to the lower bound of the worst-case weighted validation accuracy $(1 - \mathrm{WrVaEr^{UB}})$. The first column shows results for $\lambda = n \cdot 10^{-3}$, the second column for $\lambda = \lambda_{\mathrm{best}}$ by cross-validation, the third column for $\lambda = n \cdot 10^{-1.5}$, and the fourth column for $\lambda = n$.

Second, let us compare the columns in the bottom row of Figure 7. In the first column, the model provides strong performance guarantees without any sample deletion or weight change ($a = 1.0$). However, even slight deletions or small weight changes cause the theoretical guarantees to break down. In contrast, in the third and fourth columns, where $\lambda = n \cdot 10^{-1.5}, n$, the method still provides broader guarantees even with sample deletions or weight changes. As the sample deletion ratio increases, the method can provide wider guarantees, although this comes at the cost of reduced guaranteed performance. These observations reveal a trade-off between weight changes, sample deletion ratios, and guaranteed model performance. This trade-off is likely influenced by the choice of the regularization parameter. As shown in equation 19, a smaller $\lambda$ results in a larger parameter bound, while a larger $\lambda$ leads to a tighter parameter bound. This effect directly impacts the range of accuracy lower bounds that can be guaranteed.

# 6  Conclusions

In this paper, we proposed DRCS as a robust coreset selection method when the data distribution in deployment environment is uncertain. The proposed DRCS method effectively reduces data storage and model update costs in a DR learning environment. Our technical contribution is deriving an upper bound of the worst-case weighted validation error under covariate shift, and then, we perform coreset selection aimed at minimizing this upper bound. As a result, the upper bound of the test error under future uncertain covariate shift was estimated, and its theoretical guarantee was provided. Furthermore, the effectiveness of the proposed DRCS method was also demonstrated in the experiments.

We are thinking of the following possible future directions. One is the application to the regression problems or multi-class classification problems. Although the computation of the upper bound of the validation error become different, we believe that the same strategy is available. Another is the extension to *dataset distillation*, which allows not only removing samples but also newly creating samples that gives similar model parameters to that by the original dataset. Also, a part of existing methods (Munteanu et al., 2018; Tukan et al., 2020; 2021; Tolochinksy et al., 2022; Alishahi & Phillips, 2024) imply two possible future directions: One is that, since a part of coreset selection methods allows us not only to select coresets but also weight the selected samples, we consider extending our method to employ the strategy. The other is that, since these methods provide probabilistic guarantee in the performance, we consider deriving the performance guarantee of our method. Since current upper bound of the validation error in the proposed method is deterministic and therefore tends to be loose, probabilistic analysis may help this.

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

# A    Proofs of general lemmas

In this appendix, we present a collection of lemmas that are utilized in other sections of the appendix.

**Lemma A.1.** (Fenchel-Moreau theorem) *Let $f : \mathbb{R}^d \to \mathbb{R} \cup \{+\infty\}$ be a convex function. The biconjugate $f^{**}$ ($= (f^*)^*$) coincides with $f$ if $f$ is convex, proper (i.e., $\exists p \in \mathbb{R}^d : f(p) < +\infty$), and lower-semicontinuous.*

*Proof.* Refer to Section 12 of (Rockafellar, 1970) for details. □

As a specific instance of Lemma A.1, the result holds when $f$ is finite-valued ($\forall p \in \mathbb{R}^d : f(p) < +\infty$) and convex.

**Lemma A.2.** *For a convex function $f : \mathbb{R}^d \to \mathbb{R} \cup \{+\infty\}$, the following statements hold:*

- *If $f$ is proper and $\nu$-smooth, then $f^*$ is $(1/\nu)$-strongly convex.*

- *If $f$ is proper, lower-semicontinuous, and $\kappa$-strongly convex, then $f^*$ is $(1/\kappa)$-smooth.*

*Proof.* See Section X.4.2 of (Hiriart-Urruty & Lemaréchal, 1993) for further explanation. □

**Corollary A.3.** *Assume that the regularization function $\rho$ in equation 7 is $\kappa$-strongly convex, a condition necessary for applying DRCS. Then, by Lemma A.2, $\rho^*$ must be $(1/\kappa)$-smooth. This ensures that $\rho^*$ cannot take infinite values in this context.*

**Lemma A.4.** *Let $f : \mathbb{R}^d \to \mathbb{R} \cup \{+\infty\}$ be a $\kappa$-strongly convex function, and let $p^* = \mathrm{argmin}_{p \in \mathbb{R}^d} f(p)$ be its minimizer. For any $p \in \mathbb{R}^d$, the following inequality holds:*

$$\|p - p^*\|_2 \leq \sqrt{\frac{2}{\kappa}[f(p) - f(p^*)]}.$$

*Proof.* Refer to (Ndiaye et al., 2015) for a detailed proof. □

**Lemma A.5.** *For any vectors $a, c \in \mathbb{R}^n$ and a positive scalar $S > 0$, the following holds:*

$$\min_{p \in \mathbb{R}^n : \|p - c\|_2 \leq S} a^\top p = a^\top c - S\|a\|_2,$$

$$\max_{p \in \mathbb{R}^n : \|p - c\|_2 \leq S} a^\top p = a^\top c + S\|a\|_2.$$

*Proof.* Using the Cauchy-Schwarz inequality, we derive:

$$-\|a\|_2\|p - c\|_2 \leq a^\top(p - c) \leq \|a\|_2\|p - c\|_2.$$

The first inequality becomes an equality if $\exists \omega > 0 : a = -\omega(p - c)$, and the second inequality becomes an equality if $\exists \omega' > 0 : a = \omega'(p - c)$.

Since $\|p - c\|_2 \leq S$, we also have:

$$-S\|a\|_2 \leq a^\top(p - c) \leq S\|a\|_2,$$

with equality when $\|p - c\|_2 = S$.

The optimal $p$ satisfying these conditions is:

- $p = c - (S/\|a\|_2)a$ for the minimum, and

- $p = c + (S/\|a\|_2)a$ for the maximum.

Thus, the minimum and maximum values of $a^\top(p - c)$ are $-S\|a\|_2$ and $S\|a\|_2$, respectively, completing the proof. □

# B    Proofs of Definition 3.3

## B.1    Derivation of Dual Problem by Fenchel's Duality Theorem

We follow the formulation of Fenchel's duality theorem as provided in Section 31 of Rockafellar (1970).

**Lemma B.1** (A specific form of Fenchel's duality theorem: $f, g < +\infty$). *Let $f : \mathbb{R}^n \to \mathbb{R}$ and $g : \mathbb{R}^d \to \mathbb{R}$ be convex functions, and let $A \in \mathbb{R}^{n \times d}$ be a matrix. Define*

$$\boldsymbol{p}^* := \min_{\boldsymbol{p} \in \mathbb{R}^d} [f(A\boldsymbol{p}) + g(\boldsymbol{p})], \tag{22}$$

$$\boldsymbol{u}^* := \max_{\boldsymbol{u} \in \mathbb{R}^n} [-f^*(-\boldsymbol{u}) - g^*(A^\top \boldsymbol{u})]. \tag{23}$$

*According to Fenchel's duality theorem, the following equalities and conditions hold:*

$$f(A\boldsymbol{p}^*) + g(\boldsymbol{p}^*) = -f^*(-\boldsymbol{u}^*) - g^*(A^\top \boldsymbol{u}^*),$$
$$-\boldsymbol{u}^* \in \partial f(A\boldsymbol{p}^*),$$
$$\boldsymbol{p}^* \in \partial g^*(A^\top \boldsymbol{u}^*).$$

*Proof.* Introducing a dummy variable $\boldsymbol{\psi} \in \mathbb{R}^n$ and a Lagrange multiplier $\boldsymbol{u} \in \mathbb{R}^n$, the problem can be rewritten as:

$$\min_{\boldsymbol{p} \in \mathbb{R}^d} [f(A\boldsymbol{p}) + g(\boldsymbol{p})] = \max_{\boldsymbol{u} \in \mathbb{R}^n} \min_{\boldsymbol{p} \in \mathbb{R}^d, \ \boldsymbol{\psi} \in \mathbb{R}^n} [f(\boldsymbol{\psi}) + g(\boldsymbol{p}) - \boldsymbol{u}^\top (A\boldsymbol{p} - \boldsymbol{\psi})] \tag{24}$$

$$= -\min_{\boldsymbol{u} \in \mathbb{R}^n} \max_{\boldsymbol{p} \in \mathbb{R}^d, \ \boldsymbol{\psi} \in \mathbb{R}^n} [-f(\boldsymbol{\psi}) - g(\boldsymbol{p}) + \boldsymbol{u}^\top (A\boldsymbol{p} - \boldsymbol{\psi})]$$

$$= -\min_{\boldsymbol{u} \in \mathbb{R}^n} \max_{\boldsymbol{p} \in \mathbb{R}^d, \ \boldsymbol{\psi} \in \mathbb{R}^n} [\{(-\boldsymbol{u})^\top \boldsymbol{\psi} - f(\boldsymbol{\psi})\} + \{(A^\top \boldsymbol{u})^\top \boldsymbol{p} - g(\boldsymbol{p})\}]$$

$$= -\min_{\boldsymbol{u} \in \mathbb{R}^n} [f^*(-\boldsymbol{u}) + g^*(A^\top \boldsymbol{u})] = \max_{\boldsymbol{u} \in \mathbb{R}^n} [-f^*(-\boldsymbol{u}) - g^*(A^\top \boldsymbol{u})]. \tag{25}$$

The optimal solutions $\boldsymbol{p}^*$, $\boldsymbol{\psi}^*$, and $\boldsymbol{u}^*$ must satisfy the following conditions based on the optimality criteria:

$$A\boldsymbol{p}^* = \boldsymbol{\psi}^*, \quad A^\top \boldsymbol{u}^* \in \partial g(\boldsymbol{p}^*), \quad -\boldsymbol{u}^* \in \partial f(\boldsymbol{\psi}^*) = \partial f(A\boldsymbol{p}^*).$$

Similarly, introducing a dummy variable $\boldsymbol{\phi} \in \mathbb{R}^d$ and a Lagrange multiplier $\boldsymbol{p} \in \mathbb{R}^d$, the dual problem can be reformulated as:

$$\max_{\boldsymbol{u} \in \mathbb{R}^n} [-f^*(-\boldsymbol{u}) - g^*(A^\top \boldsymbol{u})] = \min_{\boldsymbol{p} \in \mathbb{R}^d} \max_{\boldsymbol{u} \in \mathbb{R}^n, \boldsymbol{\phi} \in \mathbb{R}^d} [-f^*(-\boldsymbol{u}) - g^*(\boldsymbol{\phi}) - \boldsymbol{p}^\top (A^\top \boldsymbol{u} - \boldsymbol{\phi})] \tag{26}$$

$$= \min_{\boldsymbol{p} \in \mathbb{R}^d} \max_{\boldsymbol{u} \in \mathbb{R}^n, \boldsymbol{\phi} \in \mathbb{R}^d} [\{(A\boldsymbol{p})^\top (-\boldsymbol{u}) - f^*(-\boldsymbol{u})\} + \{\boldsymbol{p}^\top \boldsymbol{\phi} - g^*(\boldsymbol{\phi})\}]$$

$$= \min_{\boldsymbol{p} \in \mathbb{R}^d} [f^{**}(A\boldsymbol{p}) + g^{**}(\boldsymbol{p})] = \min_{\boldsymbol{p} \in \mathbb{R}^d} [f(A\boldsymbol{p}) + g(\boldsymbol{p})], \quad (\because \text{ Lemma A.1})$$

The optimal solutions $\boldsymbol{u}^*$, $\boldsymbol{\phi}^*$, and $\boldsymbol{p}^*$ for the dual problem satisfy:

$$A^\top \boldsymbol{u}^* = \boldsymbol{\phi}^*, \quad \boldsymbol{p}^* \in \partial g^*(\boldsymbol{\phi}^*) = \partial g^*(A^\top \boldsymbol{u}^*), \quad A\boldsymbol{p}^* \in \partial f(-\boldsymbol{u}^*).$$

$\square$

**Lemma B.2** (Dual problem of weighted regularized empirical risk minimization). *Let us consider linear predictions including the kernel method. For the minimization problem*

$$\boldsymbol{\beta}^*_{\boldsymbol{v},\boldsymbol{w}} := \arg\min_{\boldsymbol{\beta} \in \mathbb{R}^k} P_{\boldsymbol{v},\boldsymbol{w}}(\boldsymbol{\beta}), \quad where \quad P_{\boldsymbol{v},\boldsymbol{w}}(\boldsymbol{\beta}) = \frac{1}{\sum_{i \in [n]} v_i w_i} \sum_{i \in [n]} v_i w_i \ell(y_i, \boldsymbol{\beta}^\top \boldsymbol{\phi}(\boldsymbol{x}_i)) + \rho(\boldsymbol{\beta}).$$

*The corresponding dual problem, obtained by applying Fenchel's duality theorem (Lemma B.1), is given by*

$$\boldsymbol{\alpha}^*_{\boldsymbol{v},\boldsymbol{w}} := \arg\max_{\boldsymbol{\alpha}\in\mathbb{R}^n} D_{\boldsymbol{w}}(\boldsymbol{\alpha}),$$

$$\text{where} \quad D_{\boldsymbol{v},\boldsymbol{w}}(\boldsymbol{\alpha}) = -\frac{1}{\sum_{i\in[n]} v_i w_i} \sum_{i\in[n]} v_i w_i \ell^*(-\alpha_i) - \rho^*\left(\frac{1}{\sum_{i\in[n]} v_i w_i}(\operatorname{diag}(\boldsymbol{v}\otimes\boldsymbol{w}\otimes\boldsymbol{y})\Phi)^\top\boldsymbol{\alpha}\right).$$

$$((16)\ \text{restated})$$

*Here, let us denote* $\Phi := [\boldsymbol{\phi}(\boldsymbol{x}_1) \quad \boldsymbol{\phi}(\boldsymbol{x}_2)\dots\boldsymbol{\phi}(\boldsymbol{x}_n)]^\top \in \mathbb{R}^{n\times k}$. *The primal and dual solutions,* $\boldsymbol{\beta}^*_{\boldsymbol{v},\boldsymbol{w}}$ *and* $\boldsymbol{\alpha}^*_{\boldsymbol{v},\boldsymbol{w}}$, *satisfy the following conditions:*

$$P_{\boldsymbol{v},\boldsymbol{w}}(\boldsymbol{\beta}^*_{\boldsymbol{v},\boldsymbol{w}}) = D_{\boldsymbol{v},\boldsymbol{w}}(\boldsymbol{\alpha}^*_{\boldsymbol{v},\boldsymbol{w}}),$$
$$\boldsymbol{\beta}^*_{\boldsymbol{v},\boldsymbol{w}} \in \partial\rho^*((\operatorname{diag}(\boldsymbol{v}\otimes\boldsymbol{w}\otimes\boldsymbol{y})\Phi)^\top\boldsymbol{\alpha}^*_{\boldsymbol{v},\boldsymbol{w}}),$$
$$\forall i\in[n]: \quad -\alpha^*_{\boldsymbol{v},\boldsymbol{w},i} \in \partial\ell(y_i, \boldsymbol{\beta}^{*\top}_{\boldsymbol{v},\boldsymbol{w}}\boldsymbol{\phi}(\boldsymbol{x}_i)).$$

*Proof.* To apply Fenchel's duality theorem, we set $f$, $g$, and $A$ in Lemma B.1 as:

$$f(\boldsymbol{u}) := \frac{1}{\sum_{i\in[n]} v_i w_i} \sum_{i\in[n]} v_i w_i \ell(u_i), \quad g(\boldsymbol{\beta}) := \rho(\boldsymbol{\beta}), \quad A := \operatorname{diag}(\boldsymbol{y})\Phi.$$

The conjugate function of $f$ is computed as:

$$f^*(\boldsymbol{u}) = \sup_{\boldsymbol{u}'\in\mathbb{R}^n} \left[\boldsymbol{u}^\top\boldsymbol{u}' - \frac{1}{E}\sum_{i\in[n]} v_i w_i \ell(u_i')\right] = \frac{1}{E}\sum_{i\in[n]} v_i w_i \ell^*\left(\frac{u_i}{v_i w_i}E\right),$$

where let us denote $E = \sum_{i\in[n]} v_i w_i$. Thus, the dual objective from equation 23 becomes:

$$-f^*(-\boldsymbol{u}) - g^*(A^\top\boldsymbol{u}) = -\frac{1}{E}\sum_{i\in[n]} v_i w_i \ell^*\left(-\frac{u_i}{v_i w_i}E\right) - \rho^*((\operatorname{diag}(\boldsymbol{y})\Phi)^\top\boldsymbol{u}).$$

Rewriting $u_i \leftarrow \frac{1}{E}v_i w_i\alpha_i$, or equivalently $\boldsymbol{u} \leftarrow \frac{1}{E}(\boldsymbol{v}\otimes\boldsymbol{w}\otimes\boldsymbol{\alpha})$, we obtain the dual problem of equation 16. The relationships between the primal and dual problems can be expressed as:

$$-\boldsymbol{u}^* \in \partial f(A\boldsymbol{p}^*) \ \Rightarrow\ -\frac{1}{E}(\boldsymbol{v}\otimes\boldsymbol{w}\otimes\boldsymbol{\alpha}^*_{\boldsymbol{v},\boldsymbol{w}}) \in \partial f(\operatorname{diag}(\boldsymbol{y})\Phi\boldsymbol{\beta}^*_{\boldsymbol{v},\boldsymbol{w}})$$
$$\Rightarrow -\frac{1}{E}v_i w_i\alpha^*_{\boldsymbol{v},\boldsymbol{w},i} \in \frac{1}{E}v_i w_i\partial\ell(y_i, \boldsymbol{\beta}^{*\top}_{\boldsymbol{v},\boldsymbol{w}}\boldsymbol{\phi}(\boldsymbol{x}_i)) \Rightarrow -\alpha^*_{\boldsymbol{v},\boldsymbol{w},i} \in \partial\ell(y_i, \boldsymbol{\beta}^{*\top}_{\boldsymbol{v},\boldsymbol{w}}\boldsymbol{\phi}(\boldsymbol{x}_i)),$$
$$\boldsymbol{p}^* \in \partial g^*(A^\top\boldsymbol{u}^*) \ \Rightarrow\ \boldsymbol{\beta}^*_{\boldsymbol{v},\boldsymbol{w}} \in \partial g^*((\operatorname{diag}(\boldsymbol{y})\Phi)^\top(\boldsymbol{v}\otimes\boldsymbol{w}\otimes\boldsymbol{\alpha}^*_{\boldsymbol{v},\boldsymbol{w}})) = \partial\rho^*((\operatorname{diag}(\boldsymbol{v}\otimes\boldsymbol{w}\otimes\boldsymbol{y})\Phi)^\top\boldsymbol{\alpha}^*_{\boldsymbol{v},\boldsymbol{w}}).$$

$$\square$$

## C Proofs and additional discussions of Section 3

### C.1 Proof of Theorem 3.4

In this appendix, we provide the complete proof of theorem 3.4.

**Theorem C.1.** *Assume that $\rho$ in the primal objective function $P_{\boldsymbol{1}_n,\boldsymbol{1}_n}$ is $\mu$-strongly convex with respect to $\boldsymbol{\beta}$. Let us denote the optimal primal and dual solutions for the entire training set (i.e., $v_i = 1\ \forall i\in[n]$) with uniform weights (i.e., $w_i = 1\ \forall i\in[n]$) as*

$$\boldsymbol{\beta}^*_{\boldsymbol{1}_n,\boldsymbol{1}_n} = \arg\min_{\boldsymbol{\beta}\in\mathbb{R}^k} P_{\boldsymbol{1}_n,\boldsymbol{1}_n}(\boldsymbol{\beta}) \quad and \quad \boldsymbol{\alpha}^*_{\boldsymbol{1}_n,\boldsymbol{1}_n} = \arg\max_{\boldsymbol{\alpha}\in\mathbb{R}^n} D_{\boldsymbol{1}_n,\boldsymbol{1}_n}(\boldsymbol{\alpha}),$$

*respectively. Then, an upper bound of the worst-case weighted validation error is written as*

$$\mathrm{WrVaEr}(\boldsymbol{v}) \leq \mathrm{WrVaEr}^{\mathrm{UB}}(\boldsymbol{v}) = 1 - \left( \mathbf{1}_{n'}^{\top} \boldsymbol{\zeta}(\boldsymbol{v}) - Q \sqrt{\|\boldsymbol{\zeta}(\boldsymbol{v})\|_2^2 - \frac{(\mathbf{1}_{n'}^{\top} \boldsymbol{\zeta}(\boldsymbol{v}))^2}{n'}} \right) \frac{1}{n'}, \qquad \text{((18) restated)}$$

*where,*

$$\boldsymbol{\zeta}(\boldsymbol{v}) \in \{0, 1\}^{n'}; \quad \zeta_i(\boldsymbol{v}) = I \left\{ y_i' \boldsymbol{\beta}_{\mathbf{1}_n, \mathbf{1}_n}^{*\top} \boldsymbol{\phi}(\boldsymbol{x}_i') - \|\boldsymbol{\phi}(\boldsymbol{x}_i')\|_2 \sqrt{\frac{2}{\lambda} \max_{\boldsymbol{w} \in \mathcal{W}} \mathrm{DG}(\boldsymbol{v}, \boldsymbol{w})} > 0 \right\}, \qquad \text{((19) restated)}$$

$$\mathrm{DG}(\boldsymbol{v}, \boldsymbol{w}) := P_{\boldsymbol{v}, \boldsymbol{w}}(\boldsymbol{\beta}_{\mathbf{1}_n, \mathbf{1}_n}^*) - D_{\boldsymbol{v}, \boldsymbol{w}}(\boldsymbol{\alpha}_{\mathbf{1}_n, \mathbf{1}_n}^*). \qquad \text{((20) restated)}$$

*Here, the quantity* DG *is called the* duality gap *and it plays a main role of the upper bound of the validation error.*

*Especially, if we use L2-regularization* $\rho(\beta) := \frac{\lambda}{2} \|\boldsymbol{\beta}\|_2^2$, $\mathrm{DG}(\boldsymbol{v}, \boldsymbol{w})$, *the maximization* $\max_{\boldsymbol{w} \in \mathcal{W}} \mathrm{DG}(\boldsymbol{v}, \boldsymbol{w})$ *can be algorithmically computed.*

This can be proved as follows.

First, we derive the bound of model parameters.

**Lemma C.2.** *Without loss of generality, we assume that the last* $(n^{\mathrm{old}} - n^{\mathrm{new}})$ *instances are removed* $(n^{\mathrm{new}} < n^{\mathrm{old}}$, *and* $\boldsymbol{x}_{i:}^{\mathrm{old}} = \boldsymbol{x}_{i:}^{\mathrm{new}} \quad \forall i \in [n^{\mathrm{new}}])$. *Let* $P_{\boldsymbol{w}}$ *be* $\mu$-*strongly convex, and suppose* $\boldsymbol{\beta}_{\mathbf{1}_n, \mathbf{1}_n}^* \in \mathbb{R}^k$ *and* $\boldsymbol{\alpha}_{\mathbf{1}_n, \mathbf{1}_n}^* \in \mathbb{R}^n$ *are given. Then, we can assure that the following* $\mathcal{B}_{\boldsymbol{v}, \boldsymbol{w}} \subset \mathbb{R}^k$ *must satisfy* $\boldsymbol{\beta}_{\boldsymbol{v}, \boldsymbol{w}}^* \in \mathcal{B}_{\boldsymbol{v}, \boldsymbol{w}}$:

$$\mathcal{B}_{\boldsymbol{v}, \boldsymbol{w}} := \left\{ \boldsymbol{\beta} \in \mathbb{R}^k \mid \|\boldsymbol{\beta} - \boldsymbol{\beta}_{\mathbf{1}_n, \mathbf{1}_n}^*\|_2 \leq R := \sqrt{\frac{2}{\lambda} \mathrm{DG}(\boldsymbol{v}, \boldsymbol{w})} \right\}. \qquad (27)$$

*Proof.* See Section 4.2.1 of (Hanada et al., 2023) $\qquad \square$

Here, DG takes various values depending on the possible weights $\boldsymbol{w}$ and coreset vector $\boldsymbol{v}$.

Next, we calculate a bound of the weighted validation error using $\mathcal{B}_{\boldsymbol{v}, \boldsymbol{w}}$.

**Theorem C.3.** *The range of the weighted validation error is derived using the bound of model parameters after retraining,* $\mathcal{B}_{\boldsymbol{v}, \boldsymbol{w}} \subset \mathbb{R}^k$, *where* $\boldsymbol{\beta}_{\boldsymbol{v}, \boldsymbol{w}}^* \in \mathcal{B}_{\boldsymbol{v}, \boldsymbol{w}}$, *as follows:*

$$\frac{n_{\mathrm{surelyincorrect}}^{(\boldsymbol{w}')}}{\sum_{i \in [n']} w_i'} \leq \mathrm{VaEr} \leq \frac{n_{\mathrm{surelyincorrect}}^{(\boldsymbol{w}')} + n_{\mathrm{unknown}}^{(\boldsymbol{w}')}}{\sum_{i \in [n']} w_i'} = \frac{n' - n_{\mathrm{surelycorrect}}^{(\boldsymbol{w}')}}{\sum_{i \in [n']} w_i'}, \qquad (28)$$

$$where \quad n_{\mathrm{surelycorrect}}^{(\boldsymbol{w}')} = \sum_{i \in [n']} w_i' I \left[ \min_{\boldsymbol{\beta} \in \mathcal{B}_{\boldsymbol{v}, \boldsymbol{w}}} y_i' \boldsymbol{\beta}^{\top} \boldsymbol{\phi}(\boldsymbol{x}_i') > 0 \right], \qquad (29)$$

$$n_{\mathrm{surelyincorrect}}^{(\boldsymbol{w}')} = \sum_{i \in [n']} w_i' I \left[ \max_{\boldsymbol{\beta} \in \mathcal{B}_{\boldsymbol{v}, \boldsymbol{w}}} y_i' \boldsymbol{\beta}^{\top} \boldsymbol{\phi}(\boldsymbol{x}_i') < 0 \right], \qquad (30)$$

$$n_{\mathrm{unknown}}^{(\boldsymbol{w}')} = \sum_{i \in [n']} w_i' I \left[ \min_{\boldsymbol{\beta} \in \mathcal{B}_{\boldsymbol{v}, \boldsymbol{w}}} y_i' \boldsymbol{\beta}^{\top} \boldsymbol{\phi}(\boldsymbol{x}_i') < 0, \quad \max_{\boldsymbol{\beta} \in \mathcal{B}_{\boldsymbol{v}, \boldsymbol{w}}} y_i' \boldsymbol{\beta}^{\top} \boldsymbol{\phi}(\boldsymbol{x}_i') > 0 \right], \qquad (31)$$

$$n' = n_{\mathrm{surelycorrect}}^{(\boldsymbol{w}')} + n_{\mathrm{surelyincorrect}}^{(\boldsymbol{w}')} + n_{\mathrm{unknown}}^{(\boldsymbol{w}')} = \mathbf{1}_{n'}^{\top} \boldsymbol{w}'. \qquad (32)$$

Here, equation 30 indicates that if the bound of model parameters after retraining is known, some of the test instances can be predicted. Moreover, equation 31 indicates that knowing only the bound of model parameters is insufficient to determine the correctness of classification. The interpretation of equation 28 is as follows. If $n_{\mathrm{unknown}}^{(\boldsymbol{w}')}$ is assumed to be entirely misclassified, the validation error reaches its maximum possible

value, resulting in the rightmost side of the inequality. Conversely, if all are assumed to be correctly classified, the validation error reaches its minimum possible value, resulting in the leftmost side of the inequality. That is, the range of the worst-case test accuracy after retraining can be expressed as equation 28.

Next, using $\mathcal{B}_{\boldsymbol{v},\boldsymbol{w}}$, we describe the method for computing an upper and lower bounds of the linear score $y_i' \boldsymbol{\beta}^\top \phi(\boldsymbol{x}_i')$, which is necessary for the classification in equation 30 and equation 31. In general, when $\mathcal{B}_{\boldsymbol{v},\boldsymbol{w}}$ is represented as a hypersphere, these bounds can be explicitly obtained.

**Lemma C.4.** *If $\mathcal{B}_{\boldsymbol{v},\boldsymbol{w}}$ is given as a hypersphere of radius $R \in \mathbb{R}_{\geq 0}$ centered at the original model parameter $\boldsymbol{\beta}^*_{\mathbf{1}_n, \mathbf{1}_n}$, $\left( \mathcal{B}_{\boldsymbol{v},\boldsymbol{w}} := \left\{ \boldsymbol{\beta} \in \mathbb{R}^k \mid \|\boldsymbol{\beta} - \boldsymbol{\beta}^*_{\mathbf{1}_n, \mathbf{1}_n}\|_2 \leq R \right\} \right)$, an upper and lower bounds of the linear score $y_i' \boldsymbol{\beta}^\top \boldsymbol{x}_i'$ can be analytically calculated as*

$$\min_{\boldsymbol{\beta} \in \mathcal{B}_{\boldsymbol{v},\boldsymbol{w}}} y_i' \boldsymbol{\beta}^\top \phi(\boldsymbol{x}_i') = y_i' \boldsymbol{\beta}^{*\top}_{\mathbf{1}_n, \mathbf{1}_n} \phi(\boldsymbol{x}_i') - \|y_i' \phi(\boldsymbol{x}_i')\|_2 R, \tag{33}$$

$$\max_{\boldsymbol{\beta} \in \mathcal{B}_{\boldsymbol{v},\boldsymbol{w}}} y_i' \boldsymbol{\beta}^\top \phi(\boldsymbol{x}_i') = y_i' \boldsymbol{\beta}^{*\top}_{\mathbf{1}_n, \mathbf{1}_n} \phi(\boldsymbol{x}_i') + \|y_i' \phi(\boldsymbol{x}_i')\|_2 R. \tag{34}$$

The proof is shown in Lemma A.5. In equation 33 and equation 34, these show that an upper and lower bounds of the linear score $y_i' \boldsymbol{\beta}^\top \phi(\boldsymbol{x}_i')$ depend on the value of $R$. As $R$ increases, a lower bound is more likely to take negative values, while an upper bound is more likely to take positive values.

**Corollary C.5.** *If the bound of model parameters after retraining, $\mathcal{B}_{\boldsymbol{v},\boldsymbol{w}} \subset \mathbb{R}^k$, is specifically represented as a hypersphere with an L2 norm radius $R$, maximizing an upper bound of the worst-case weighted validation error, as shown in equation 28, is equivalent to maximizing $R$.*

Finally, considering the worst-case weighted validation error, the optimization of the weights $\boldsymbol{w}'$ assigned to the validation instances is performed.

**Lemma C.6.** *Assume that the range of validation weights $\boldsymbol{w}'$ is an L2-norm hypersphere, defined as $\mathcal{W}' := \left\{ \boldsymbol{w}' \mid \|\boldsymbol{w}' - \mathbf{1}_{n'}\|_2 \leq Q \right\}$ $(Q > 0)$, and that the sum of the weights is constant. In this case, the maximization problem for the validation weights that results in an upper bound of the weighted validation error can be formulated as follows:*

$$\max_{\boldsymbol{w}' \in \mathcal{W}'} \frac{n' - n^{(\boldsymbol{w}')}_{\text{surelycorrect}}}{\sum_{i \in [n']} w_i'}, \quad where \quad n^{(\boldsymbol{w}')}_{\text{surelycorrect}} = \boldsymbol{\zeta}(\boldsymbol{v},\boldsymbol{w})^\top \boldsymbol{w}', \quad \zeta_i = I\left[ \min_{\boldsymbol{\beta} \in \mathcal{B}_{\boldsymbol{v},\boldsymbol{w}}} y_i' \boldsymbol{\beta}^\top \phi(\boldsymbol{x}_i') > 0 \right], \tag{35}$$

$$subject\ to \quad \|\boldsymbol{w} - \mathbf{1}_{n'}\|_2 \leq Q, \quad \sum_{i \in [n']} w_i' = n', \tag{36}$$

*and the maximization of this problem can be analytically computed as follows:*

$$\max_{\boldsymbol{w}' \in \mathcal{W}'} \frac{n' - n^{(\boldsymbol{w}')}_{\text{surelycorrect}}}{\sum_{i \in [n']} w_i'} = 1 - \left( \mathbf{1}_{n'}^\top \boldsymbol{\zeta}(\boldsymbol{v},\boldsymbol{w}) - Q\sqrt{\|\boldsymbol{\zeta}(\boldsymbol{v},\boldsymbol{w})\|_2^2 - \frac{\left( \mathbf{1}_{n'}^\top \boldsymbol{\zeta}(\boldsymbol{v},\boldsymbol{w}) \right)^2}{n'}} \right) \frac{1}{n'}. \tag{37}$$

The proof is shown in Appendix C.2. Here, $\boldsymbol{\zeta}$ is a function of $\boldsymbol{v}$ and $\boldsymbol{w}$. From equation 33 and equation 35, maximizing $R$ with respect to $\boldsymbol{w}$ allows us to maximize an upper bound of the worst-case weighted validation error. This is expressed as

$$\zeta_i(\boldsymbol{v}) = I\left\{ y_i' \boldsymbol{\beta}^{*\top}_{\mathbf{1}_n, \mathbf{1}_n} \phi(\boldsymbol{x}_i') - \|\phi(\boldsymbol{x}_i')\|_2 R > 0 \right\}, \tag{38}$$

$$where\ R = \sqrt{\frac{2}{\lambda} \max_{\boldsymbol{w} \in \mathcal{W}} \text{DG}(\boldsymbol{v},\boldsymbol{w})}. \tag{39}$$

Therefore, an upper bound of the worst-case weighted validation error is written as

$$\text{WrVaEr}(\boldsymbol{v}) \leq 1 - \left( \mathbf{1}_{n'}^\top \boldsymbol{\zeta}(\boldsymbol{v}) - Q\sqrt{\|\boldsymbol{\zeta}(\boldsymbol{v})\|_2^2 - \frac{\left( \mathbf{1}_{n'}^\top \boldsymbol{\zeta}(\boldsymbol{v}) \right)^2}{n'}} \right) \frac{1}{n'} \tag{40}$$

This provides a theoretical upper-bound guarantee on the worst-case test error after retraining.

## C.2   Proof of Lemma C.6

In this appendix, we provide the proof of a following constrained maximization problem:

$$\max_{\boldsymbol{w}' \in \mathcal{W}'} \frac{n' - n_{\text{surelycorrect}}^{(\boldsymbol{w}')}}{\sum\limits_{i \in [n']} w_i'}, \quad \text{subject to } \|\boldsymbol{w}' - \mathbf{1}_{n'}\|_2 = Q, \quad \mathbf{1}_{n'}^\top \boldsymbol{w}' = n', \tag{41}$$

$$\text{where} \quad n_{\text{surelycorrect}}^{(\boldsymbol{w}')} = \boldsymbol{\zeta}^\top \boldsymbol{w}' \tag{42}$$

Then, this problem can be transformed and rewritten as follows:

$$\max_{\boldsymbol{w}' \in \mathcal{W}'} \frac{n' - n_{\text{surelycorrect}}^{(\boldsymbol{w}')}}{\sum\limits_{i \in [n']} w_i'} = 1 - \frac{\min_{\boldsymbol{w}' \in \mathcal{W}'} \boldsymbol{\zeta}^\top \boldsymbol{w}'}{n'}. \tag{43}$$

Thus, we prove that this minimization problem can be solved analytically.

*Proof.* First, we write the Lagrangian function with Lagrange multiplier $\mu, \nu \in \mathbb{R}$ as:

$$L(\boldsymbol{w}', \mu, \nu) = \boldsymbol{\zeta}^\top \boldsymbol{w}' + \mu \left( \|\boldsymbol{w}' - \mathbf{1}_{n'}\|_2^2 - Q^2 \right) + \nu \left( \mathbf{1}_{n'}^\top \boldsymbol{w}' - n' \right). \tag{44}$$

Then, due to the property of Lagrange multiplier, the stationary points of equation 43 are obtained as

$$\frac{\partial L(\boldsymbol{w}', \mu, \nu)}{\partial \boldsymbol{w}'} = \boldsymbol{\zeta} + 2\mu \left( \boldsymbol{w}' - \mathbf{1}_{n'} \right) + \nu \mathbf{1}_{n'} = 0, \tag{45}$$

$$2\mu \left( \boldsymbol{w}' - \mathbf{1}_{n'} \right) = -\boldsymbol{\zeta} - \nu \mathbf{1}_{n'}. \tag{46}$$

Squaring both sides of equation 46, we get:

$$4\mu^2 \|\boldsymbol{w}' - \mathbf{1}_{n'}\|_2^2 = \|\boldsymbol{\zeta}\|_2^2 + 2\nu \boldsymbol{\zeta}^\top \mathbf{1}_{n'} + \nu^2 n'. \tag{47}$$

Multiplying both sides of equation 46 by $\mathbf{1}_{n'}^\top$, we obtain:

$$2\mu \left( \mathbf{1}_{n'}^\top \boldsymbol{w}' - n' \right) = -\mathbf{1}_{n'}^\top \boldsymbol{\zeta} - \nu \mathbf{1}_{n'}^\top \mathbf{1}_{n'},$$

$$0 = -\mathbf{1}_{n'}^\top \boldsymbol{\zeta} - \nu n',$$

$$\therefore \nu = -\frac{\mathbf{1}_{n'}^\top \boldsymbol{\zeta}}{n'}. \tag{48}$$

Substituting equation 48 into equation 47, we obtain:

$$4\mu^2 \|\boldsymbol{w}' - \mathbf{1}_{n'}\|_2^2 = \|\boldsymbol{\zeta}\|_2^2 - \frac{2}{n'} \left( \mathbf{1}_{n'}^\top \boldsymbol{\zeta} \right)^2 + \left( -\frac{\mathbf{1}_{n'}^\top \boldsymbol{\zeta}}{n'} \right)^2 n',$$

$$4\mu^2 \|\boldsymbol{w}' - \mathbf{1}_{n'}\|_2^2 = \|\boldsymbol{\zeta}\|_2^2 - \frac{\left( \mathbf{1}_{n'}^\top \boldsymbol{\zeta} \right)^2}{n'},$$

$$2\mu = \pm \frac{1}{Q} \sqrt{\|\boldsymbol{\zeta}\|_2^2 - \frac{\left( \mathbf{1}_{n'}^\top \boldsymbol{\zeta} \right)^2}{n'}}. \tag{49}$$

Substituting equation 49 into equation 45, we obtain:

$$\boldsymbol{w}' = \mathbf{1}_{n'} \pm \frac{Q}{\sqrt{\|\boldsymbol{\zeta}\|_2^2 - \frac{\left( \mathbf{1}_{n'}^\top \boldsymbol{\zeta} \right)^2}{n'}}} \left( -\boldsymbol{\zeta} + \frac{\mathbf{1}_{n'}^\top \boldsymbol{\zeta}}{n'} \mathbf{1}_{n'} \right). \tag{50}$$

Multiplying both sides of equation 50 by $\boldsymbol{\zeta}$, we obtain:

$$\boldsymbol{\zeta}^\top \boldsymbol{w}' = \mathbf{1}_{n'}^\top \boldsymbol{\zeta} \pm \frac{Q}{\sqrt{\|\boldsymbol{\zeta}\|_2^2 - \dfrac{\left(\mathbf{1}_{n'}^\top \boldsymbol{\zeta}\right)^2}{n'}}} \left( -\|\boldsymbol{\zeta}\|_2^2 + \frac{\mathbf{1}_{n'}^\top \boldsymbol{\zeta}}{n'} \boldsymbol{\zeta}^\top \mathbf{1}_{n'} \right)$$

$$= \mathbf{1}_{n'}^\top \boldsymbol{\zeta} \mp Q \sqrt{\|\boldsymbol{\zeta}\|_2^2 - \frac{\left(\mathbf{1}_{n'}^\top \boldsymbol{\zeta}\right)^2}{n'}} \tag{51}$$

Finally, we obtain the minimum value of $\boldsymbol{\zeta}^\top \boldsymbol{w}'$:

$$\min_{\boldsymbol{w}' \in \mathcal{W}'} \boldsymbol{\zeta}^\top \boldsymbol{w}' = \mathbf{1}_{n'}^\top \boldsymbol{\zeta} - Q \sqrt{\|\boldsymbol{\zeta}\|_2^2 - \frac{\left(\mathbf{1}_{n'}^\top \boldsymbol{\zeta}\right)^2}{n'}}$$

$\square$

### C.3 Use of strongly-convex regularization functions other than L2-regularization

Let us consider the use of regularization functions other than L2-regularization, assuming that the function is $\kappa$-strongly convex to satisfy the conditions for applying DRCS. For L2-regularization, the term $\rho^*\left( \frac{1}{\sum_{i\in[n]} v_i w_i} (\mathrm{diag}(\boldsymbol{v}\otimes\boldsymbol{w}\otimes\boldsymbol{y})\Phi)^\top \boldsymbol{\alpha}^*_{\mathbf{1}_n,\mathbf{1}_n} \right)$ in the duality gap equation 20 simplifies to a quadratic function with respect to $\boldsymbol{w}$. However, for other regularization functions, even if they are $\kappa$-strongly convex, the behavior can differ significantly, potentially complicating the application of DRCS.

As a famous example, consider the *elastic net* regularization Zou & Hastie (2005). With hyperparameters $\lambda > 0$ and $\lambda' > 0$, the regularization function and its convex conjugate are defined as follows:

$$\rho(\boldsymbol{\beta}) := \frac{\lambda}{2}\|\boldsymbol{\beta}\|_2^2 + \lambda'\|\boldsymbol{\beta}\|_1,$$

$$\rho^*(\boldsymbol{p}) = \frac{1}{2\lambda} \sum_{j\in[d]} \left[ \max\{0, |p_j| - \lambda'\} \right]^2 .$$

In this case, the term

$$\rho^*\left( \frac{1}{\sum_{i\in[n]} v_i w_i} (\mathrm{diag}(\boldsymbol{v}\otimes\boldsymbol{w}\otimes\boldsymbol{y})\Phi)^\top \boldsymbol{\alpha}^*_{\mathbf{1}_n,\mathbf{1}_n} \right)$$

$$= \frac{1}{2\lambda} \sum_{j\in[d]} \left[ \max\{0, \left| \frac{1}{\sum_{i\in[n]} v_i w_i} (\boldsymbol{v}\otimes\boldsymbol{w}\otimes\boldsymbol{\alpha}^*_{\mathbf{1}_n,\mathbf{1}_n}\otimes\boldsymbol{y}\otimes\Phi_{:j}) \right| - \lambda'\} \right]^2$$

requires maximization with respect to $\boldsymbol{w}$, which is nontrivial and introduces additional complexity.

Next, we discuss a sufficient condition for regularization functions that allows weighted regularized empirical risk minimization to support both kernelization and DRCS.

**Lemma C.7.** *In weighted regularized empirical risk minimization as defined in 7, the regularization function $\rho$ can support both DRCS and kernelization if it is described as:*

$$\rho(\boldsymbol{\beta}) = \frac{\kappa}{2}\|\boldsymbol{\beta}\|_2^2 + \mathcal{H}(\|\boldsymbol{\beta}\|_2),$$

*where $\mathcal{H} : \mathbb{R}_{\geq 0} \to \mathbb{R}$ is an increasing function and $\kappa$ is a positive constant.*

*Proof.* According to the *generalized representer theorem* Schölkopf et al. (2001), weighted regularized empirical risk minimization can be kernelized if the regularization function $\rho$ can be expressed in terms of a strictly increasing function $\mathcal{G} : \mathbb{R}_{\geq 0} \to \mathbb{R}$ as $\rho(\boldsymbol{\beta}) = \mathcal{G}(\|\boldsymbol{\beta}\|_2)$.

By combining this requirement with the condition for applying DRCS, which demands that $\rho$ be $\kappa$-strongly convex, we obtain the stated form of $\rho$. $\square$

### C.4 Methods for Greedy Coreset Selection

A naive greedy approach involves removing the training instance that minimizes equation 18 one at a time (this approach is referred to as `greedy approach 1`). First, the pseudocode of `greedy approach 1` for small datasets is given in Algorithm 1.

---

**Algorithm 1** Distributionally Robust Coreset Selection for Small Datasets

---

**Input:** Dataset $\mathcal{D} := \{(\boldsymbol{x}_i, y_i)\}_{i \in [n]}$, matrix $A$, vector $\boldsymbol{b}$, constant $c$
1: Initialize $\boldsymbol{v} \leftarrow \{1\}^n$
2: Set desired number of deletions, $n^{\mathrm{del}}$
3: **while** number of deletions is less than $n^{\mathrm{del}}$ **do**
4:     **for** each $i$ where $v_i = 1$ **do**
5:         Set $\boldsymbol{v}' \leftarrow \boldsymbol{v}$ and $v_i' \leftarrow 0$                                   ▷ Remove $i$-th element from $\boldsymbol{v}$
6:         Compute the duality gap when an instance is removed:

$$\mathrm{DG}_i = \max_{\boldsymbol{w} \in \mathcal{W}} \left\{ (\boldsymbol{v}' \otimes \boldsymbol{w})^\top A (\boldsymbol{v}' \otimes \boldsymbol{w}) + \boldsymbol{b}^\top (\boldsymbol{v}' \otimes \boldsymbol{w}) + c \right\}$$

7:         Store $\mathrm{DG}_i$
8:     **end for**
9:     Find the index $i^*$ corresponding to the smallest $\mathrm{DG}_i$ value
10:     Set $v_{i^*} \leftarrow 0$                         ▷ Remove the element with the smallest maximum value
11:     Update $\boldsymbol{v}$ and repeat the process
12: **end while**
13: Construct the subset $\hat{\mathcal{D}} = \{\boldsymbol{x}_i, y_i \mid \{\boldsymbol{x}_i, y_i\} \in \mathcal{D}, v_i = 1\}$;
**Output:** Selected dataset: $\hat{\mathcal{D}}$

---

Next, the pseudocode of `greedy approach 2` for large datasets is given in Algorithm 2. It performs the maximization calculation once at the beginning, and then sequentially performs instance selection.

---

**Algorithm 2** Distributionally Robust Coreset Selection for Large Datasets

---

**Input:** Dataset $\mathcal{D} := \{(\boldsymbol{x}_i, y_i)\}_{i \in [n]}$, matrix $A$, vector $\boldsymbol{b}$, constant $c$

1: Initialize $\boldsymbol{v} \leftarrow \{1\}^n$

2: Compute worst-case weight that maximize the duality gap :

$$\boldsymbol{w}^{\text{worst}} = \arg\max_{\boldsymbol{w} \in \mathcal{W}} \left\{ (\boldsymbol{v} \otimes \boldsymbol{w})^\top A(\boldsymbol{v} \otimes \boldsymbol{w}) + \boldsymbol{b}^\top (\boldsymbol{v} \otimes \boldsymbol{w}) + c \right\}$$

3: Set desired number of deletions, $n^{\text{del}}$

4: **while** number of deletions is less than $n^{\text{del}}$ **do**

5:     **for** each $i$ where $v_i = 1$ **do**

6:         Set $\boldsymbol{v'} \leftarrow \boldsymbol{v}$ and $v'_i \leftarrow 0$                           ▷ Remove $i$-th element from $\boldsymbol{v}$

7:         Calculate $\text{DG}_i \leftarrow (\boldsymbol{v'} \otimes \boldsymbol{w}^{\text{worst}})^\top A(\boldsymbol{v'} \otimes \boldsymbol{w}^{\text{worst}}) + \boldsymbol{b}^\top (\boldsymbol{v'} \otimes \boldsymbol{w}^{\text{worst}}) + c$

8:         Store $\text{DG}_i$

9:     **end for**

10:    Find the index $i^*$ corresponding to the smallest $\text{DG}_i$ value

11:    Set $v_{i^*} \leftarrow 0$                            ▷ Remove the element with the smallest maximum value

12:    Update $\boldsymbol{v}$ and repeat the process

13: **end while**

14: Construct the subset $\hat{\mathcal{D}} = \{\boldsymbol{x}_i, y_i \mid \{\boldsymbol{x}_i, y_i\} \in \mathcal{D}, v_i = 1\}$;

**Output:** Selected dataset: $\hat{\mathcal{D}}$

---

Then, the pseudocode of `greedy approach 3` for large datasets is given in Algorithm 3.

---

**Algorithm 3** Distributionally Robust Coreset Selection for Large Datasets

---

**Input:** Dataset $\mathcal{D} := \{(\boldsymbol{x}_i, y_i)\}_{i \in [n]}$, matrix $A$, vector $\boldsymbol{b}$, constant $c$

1: Initialize $\boldsymbol{v} \leftarrow \{1\}^n$

2: Compute worst-case weight that maximize the duality gap :

$$\boldsymbol{w}^{\text{worst}} = \arg\max_{\boldsymbol{w} \in \mathcal{W}} \left\{ (\boldsymbol{v} \otimes \boldsymbol{w})^\top A(\boldsymbol{v} \otimes \boldsymbol{w}) + \boldsymbol{b}^\top (\boldsymbol{v} \otimes \boldsymbol{w}) + c \right\}$$

3: Set desired number of deletions, $n^{\text{del}}$

4: **for** each $i \in [n]$ **do**

5:     Set $\boldsymbol{v'} \leftarrow \boldsymbol{v}$ and $v'_i \leftarrow 0$                        ▷ Remove $i$-th element from $\boldsymbol{v}$

6:     Calculate $\text{DG}_i \leftarrow (\boldsymbol{v'} \otimes \boldsymbol{w}^{\text{worst}})^\top A(\boldsymbol{v'} \otimes \boldsymbol{w}^{\text{worst}}) + \boldsymbol{b}^\top (\boldsymbol{v'} \otimes \boldsymbol{w}^{\text{worst}}) + c$

7:     Store $\text{DG}_i$

8: **end for**

9: Identify $n^{\text{del}}$ smallest $\text{DG}_i$'s, and set $v_i \leftarrow 0$ for these $i$'s, or $v_i \leftarrow 1$ otherwise

10: Construct the subset $\hat{\mathcal{D}} = \{\boldsymbol{x}_i, y_i \mid \{\boldsymbol{x}_i, y_i\} \in \mathcal{D}, v_i = 1\}$;

**Output:** Selected dataset: $\hat{\mathcal{D}}$

---

Here, the duality gap $\text{DG}_i$ computed within the algorithm is a quadratic convex function with respect to $\boldsymbol{w}$. As a method to solve the constrained maximization of $\text{DG}_i$ in $\mathcal{W}$, we apply method of Lagrange multiplier. In this case, since all stationary points can be computed algorithmically, maximization is achievable. For the proof, please refer to the Appendix of Hanada et al. (2024). This maximization calculation requires $O\left(n^3\right)$ time. For this reason, applying Algorithm 1, which repeatedly requires maximization calculations, to large datasets is computationally expensive. Therefore, Algorithm 2, which reduces the number of maximization computations, is also considered. Furthermore, in large datasets, the computational cost increases even in the instance selection process by $v$. In order to avoid an increase in computational cost, we adopt Algorithm 3 for large datasets in Section 5.3.

# D  Details of Experiments

## D.1  Experimental Environments and Implementation Information

We used the following computers for experiments: For experiments except for the image dataset, we run experiments on a computer with Intel Xeon Silver 4214R (2.40GHz) CPU and 64GB RAM. For experiments using the image dataset, we run experiments on a computer with Intel(R) Xeon(R) Gold 6338 (2.00GHz) CPU, NVIDIA RTX A6000 GPU and 1TB RAM.

Procedures are implemented in Python, mainly with the following libraries:

- *NumPy* (Harris et al., 2020): Matrix and vector operations

- *CVXPY* (Diamond & Boyd, 2016): Convex optimizations (training computations with weights)

- *SciPy* (Virtanen et al., 2020): Solving equations to maximize the quadratic convex function in equation 21 (by module *optimize.root_scalar*)

- *PyTorch* (Paszke et al., 2017): Defining the source neural network (which will be converted to a kernel by NTK) for image prediction

- *neural-tangents* (Novak et al., 2020): NTK

## D.2  Data and Learning Setup

The criteria of selecting datasets (Table 1) and detailed setups are as follows:

- All of the datasets are downloaded from LIBSVM dataset (Chang & Lin, 2011). We used training datasets only if test datasets are provided separately ("splice").

- In the table, the column "$d$" denotes the number of features including the intercept feature.

The choice of the regularization hyperparameter $\lambda$, based on the characteristics of the data, is as follows: We set $\lambda$ as $n$, $n \times 10^{-1.5}$, $n \times 10^{-3.0}$ and best $\lambda$ which is decided by cross-validation.

The choice of the hyperparameter in RBF kernel is fixed as follows: we set $\zeta = d * \mathbb{V}(Z)$ as suggested in `sklearn.svm.SVC` of *scikit-learn* (Pedregosa et al., 2011), where $\mathbb{V}$ denotes the elementwise sample variance.

### D.3 All Experimental Results of Section 5.2 using logistic regression model

In this appendix D.3, we show all experimental results using logistic regression model(logistic loss + L2 regularization). First, we show model performance.

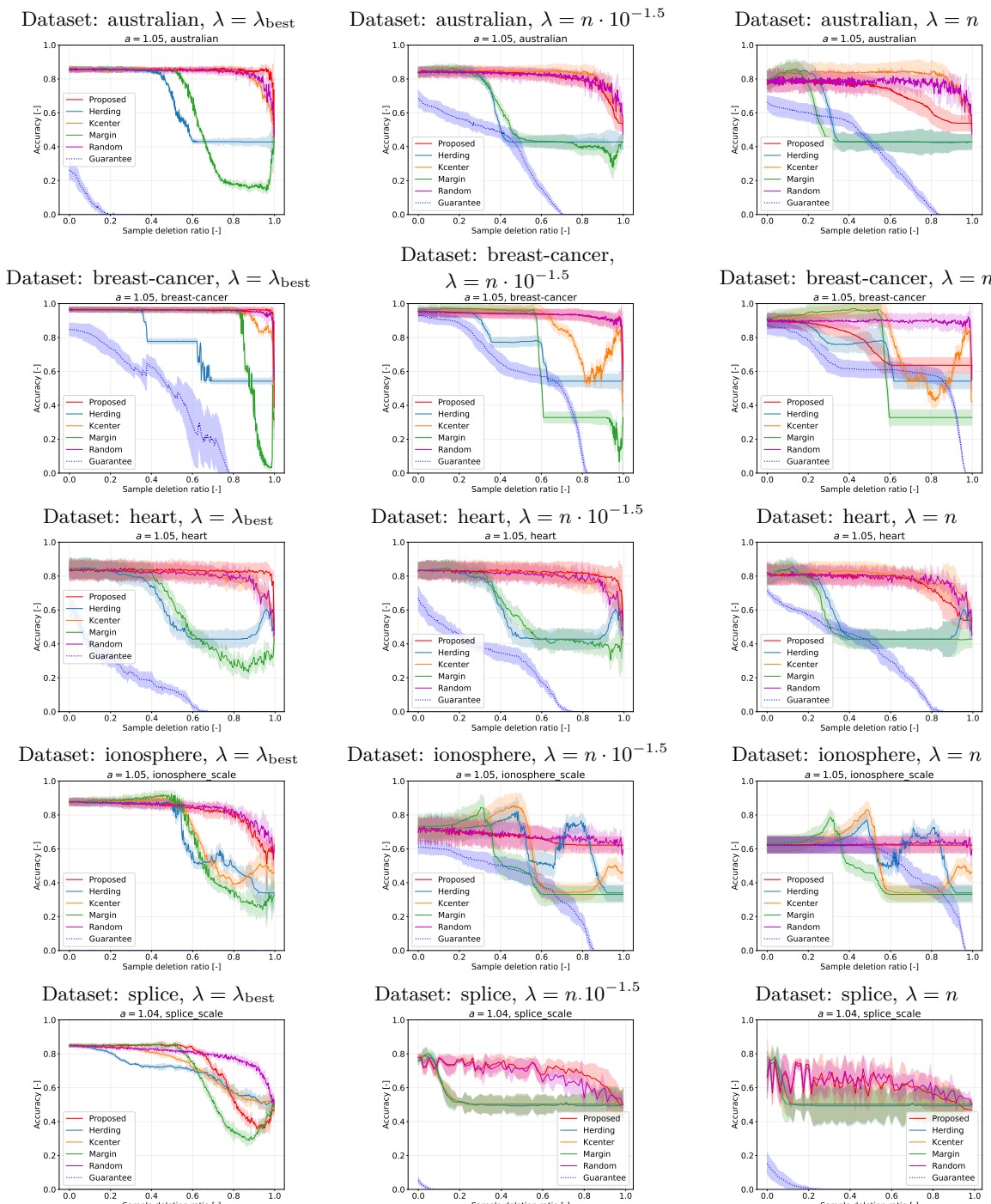

Figure 8: Model performance for RBF-kernel logistic regression models, under the settings described in Section 5 and Appendix D.2.

Next, we show a guarantee of model performance.

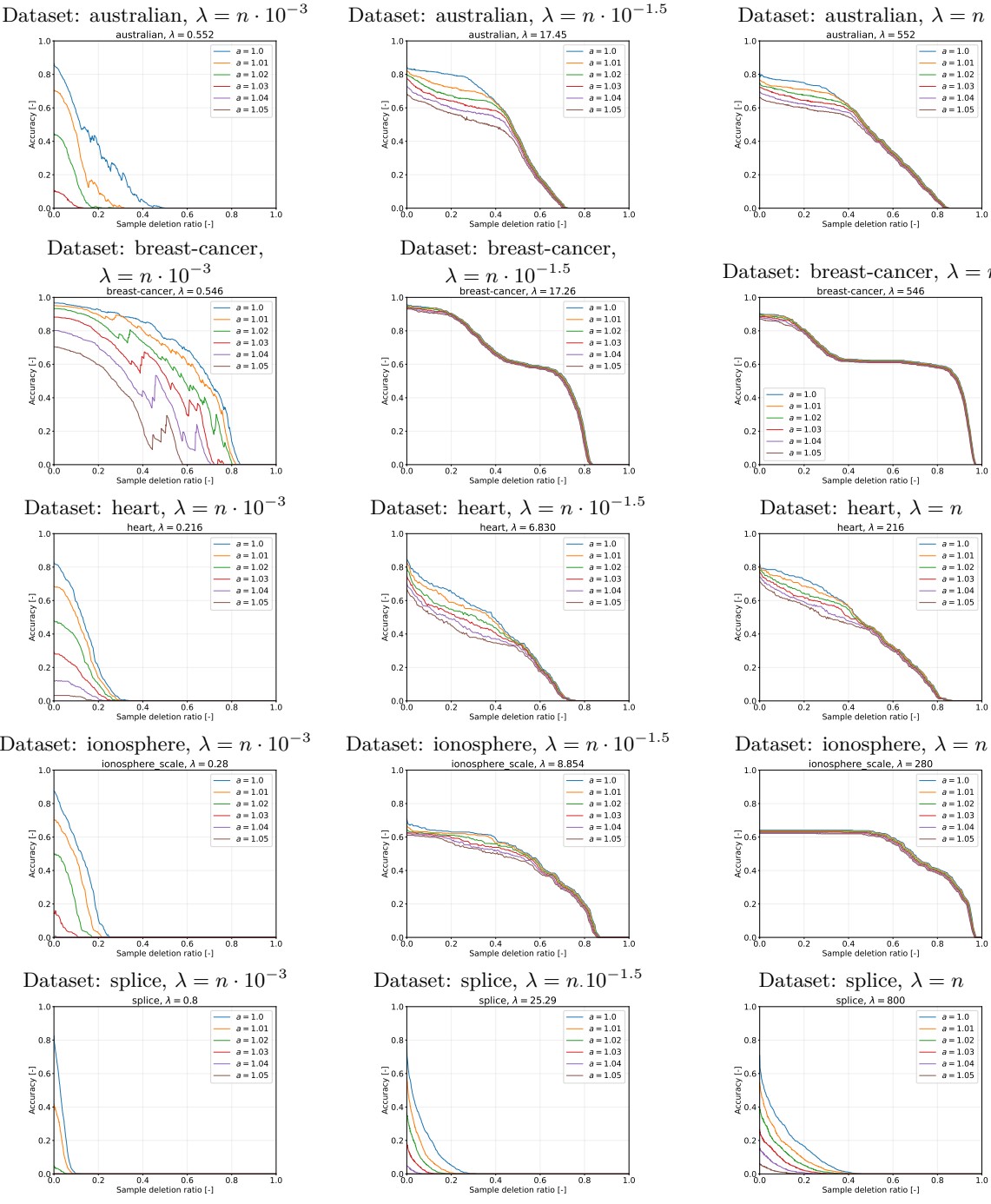

Figure 9: Model performance for RBF-kernel logistic regression models, under the settings described in Section 5 and Appendix D.2.

## D.4    All Experimental Results of Section 5.2 using support vector machine model

In this appendix D.4, we show all experimental results using support vector machine model(hinge loss + L2 regularization). First, we show model performance.

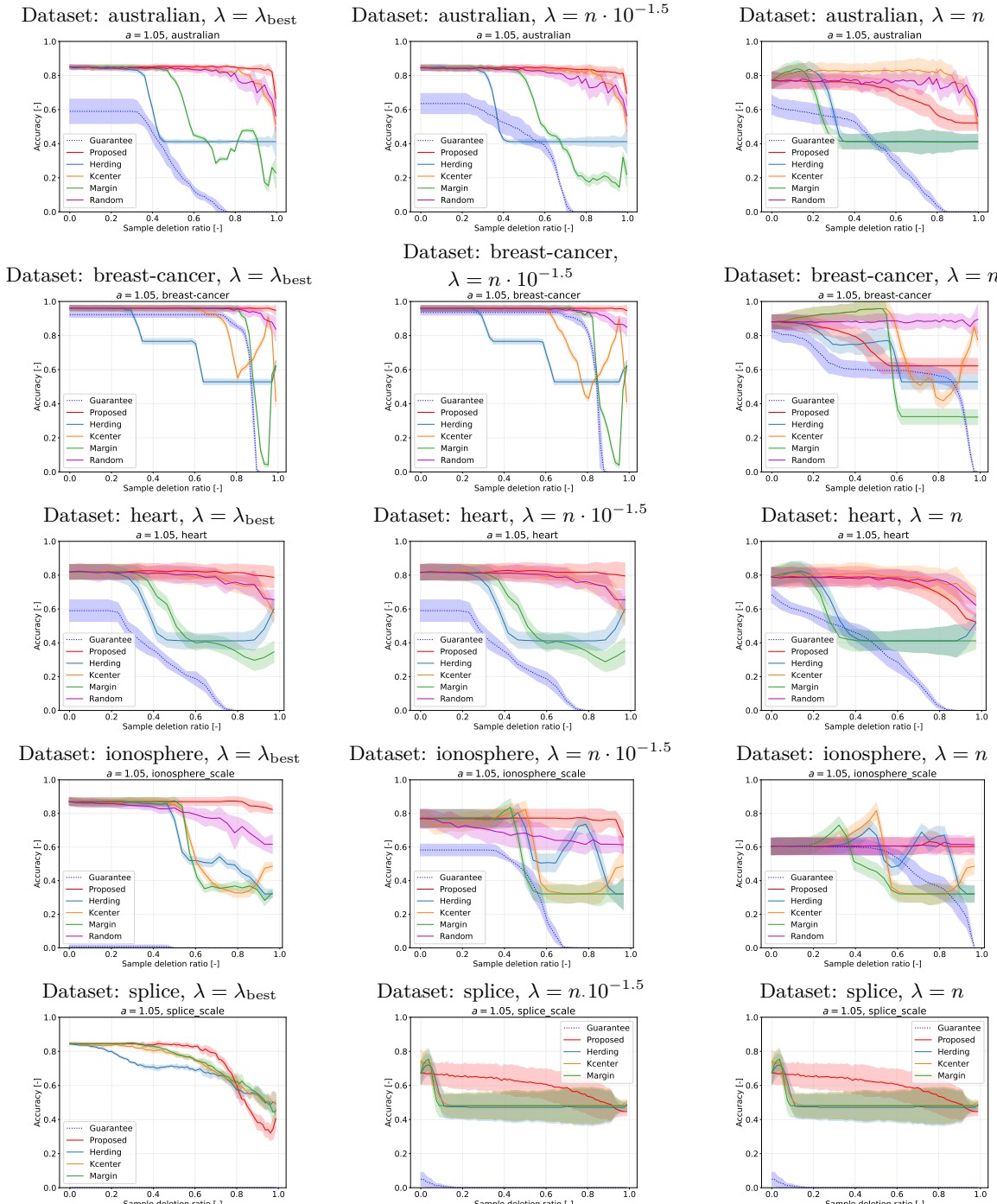

Figure 10: Model performance for RBF-kernel SVMs, under the settings described in Section 5 and Appendix D.2.

Next, we show a guarantee of model performance.

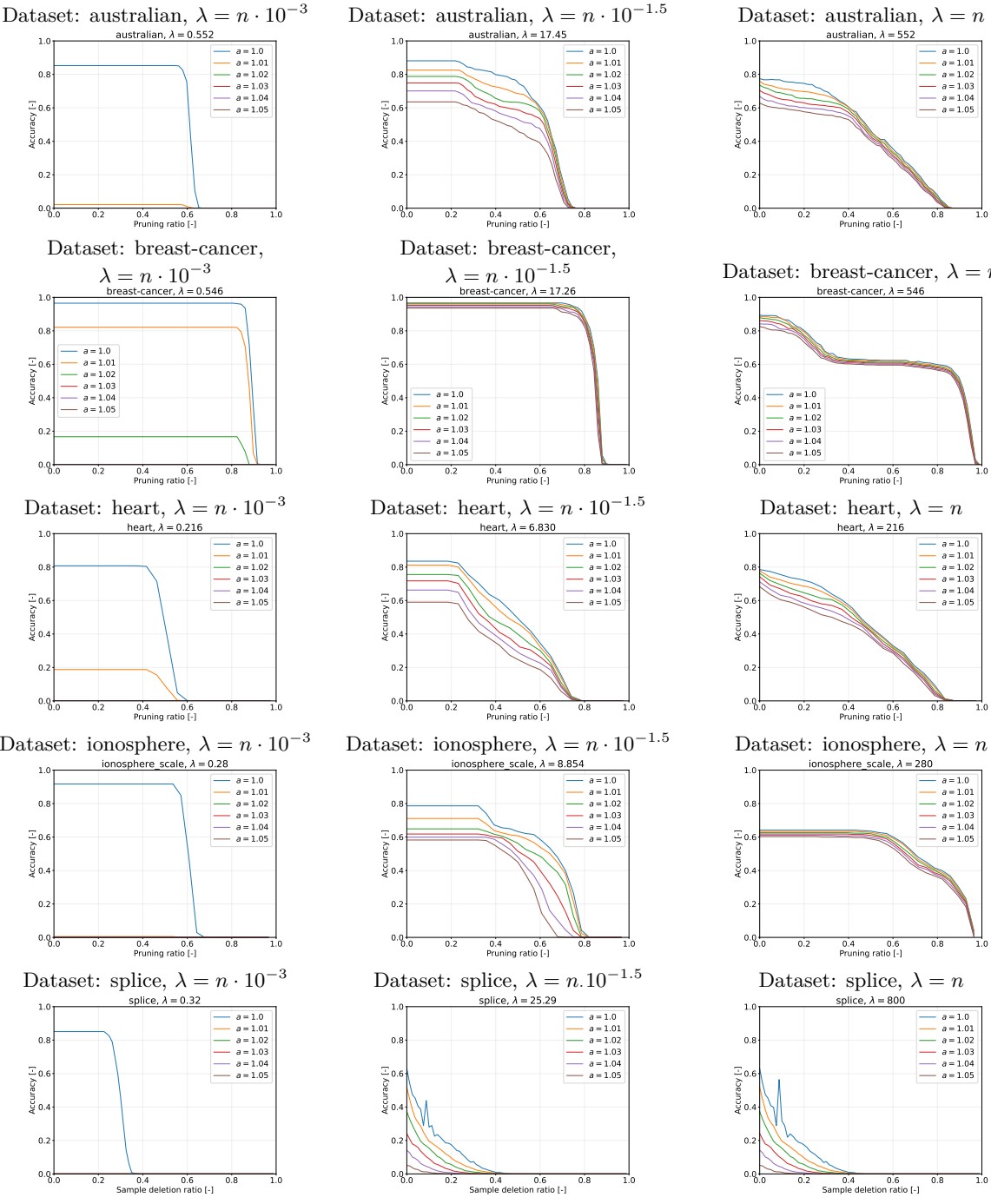

Figure 11: Model performance for RBF-kernel SVMs, under the settings described in Section 5 and Appendix D.2.

### D.5 The effectiveness of the proposed method in different models

Here, we compare the results between logistic regression in Appendix D.3 and SVM in Appendix D.4. Regarding model performance, there is no significant difference between the two models, and the trends when regularization is large or small are generally consistent for both (Figure 8 and 10). On the other hand, in terms of theoretical evaluation, there are significant differences between the two models (Figure 9 and 11). The major difference between SVM and logistic regression lies in whether the model is instance-sparse. In SVM, training instances where $\alpha^*_{\mathbf{1}_n,\mathbf{1}_n,i} = 0$ do not affect the duality gap during instance selection. Therefore, more effective instance selection is possible, as the duality gap can be suppressed further. In contrast, logistic regression is not a sparse model, and as a result, the theoretical lower bound of the worst-case weighted validation accuracy is expected to be lower than that of SVM.

### D.6 Experimental Results Using NTK in Section 5.3

We also experimented the DRCS method with NTK for an image dataset. We composed 5,000-sample binary classification dataset from "CIFAR-10" dataset by choosing from classes "airplane" and "automobile". A total of 1000 instances were sampled, and experiments were conducted using Algorithm 1. It was demonstrated that the proposed DRCS method could be approximately applied to deep learning by using NTK.

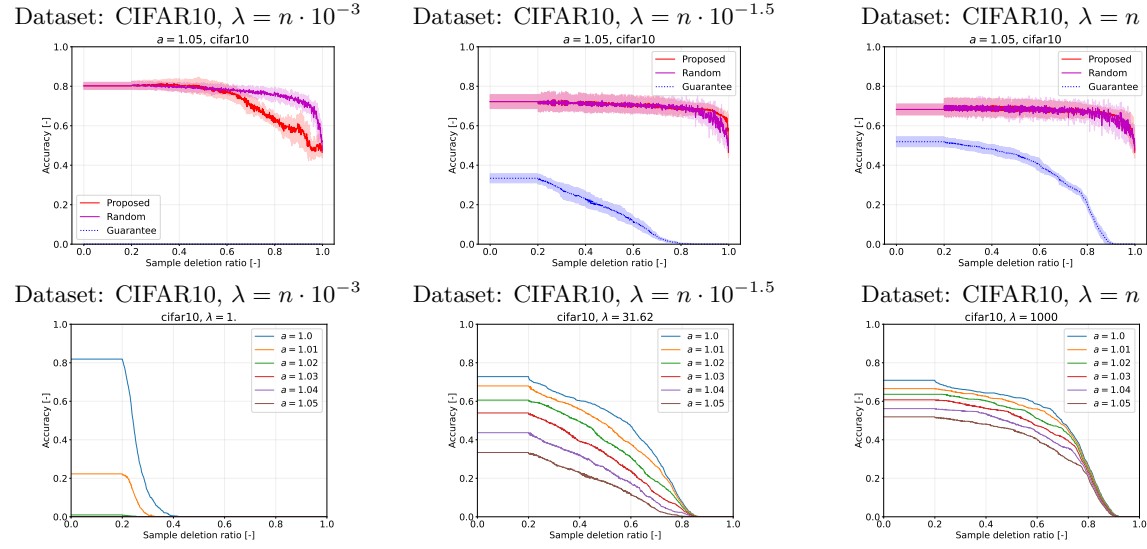

Figure 12: Model performance for NTK, under the settings described in Section 5 and Appendix D.2.

