# OpenReview forum: "Distributionally Robust Coreset Selection under Covariate Shift"
_TMLR — Accepted by TMLR_

### Review · Reviewer_g4Vy · 2025-02-27

**Summary Of Contributions:**

The paper suggests the first coreset that handles distribution shift, specifically covariate shift tailored for machine learning applications that use strongly convex functions.

**Audience:**

Yes

**Broader Impact Concerns:**

I have no concerns about the ethical implications of the work.

**Claims And Evidence:**

No

**Requested Changes:**

1) Equation 11, should $\mathcal{W}$ be $\mathcal{W}^\prime$?
2) What is $\ell^\ast$ is at Equation 16?
3) What is $\rho^\ast$ is at Equation 16? Is it related to the conjugate convex of $\rho$?
4) At the second item list of the sketch of proof of Theorem 3.3 (section 3.1.2), what is assumption 1?
5) what is the motivation behind setting $S:= \sqrt{n^\+}\left| a-1 \right|$?
6) In Figure 5, setting a to 1 means that $S=0$?
7) Regarding section 5.3, I didn't quite follow the experiment. Did you train the model on the entire data, then take the features from ResNet50, choose a subset as the coreset, and then retrain the entire model on the subset? Why not simply use the pretrained model of ResNet50, and then train on the subset similar to either [5] or [6] for example?
8) Page 12, did you mean to write "calculation" instead of "calcuration"?
9) Page 17, In Lemma A.1., did you mean to write $f^\ast$ instead of $f^{\ast\ast}$?
10) The paper uses $\otimes$, which I assume is the element-wise multiplication. If that's the case, please make sure to explicitly state this.
11) Page 19, what should be instead of the "equation ??"?
12) In the proof of Lemma C.6, is $\sum_{i \in [n^\prime]} w_i^\prime = n^\prime$? Please connect between equation 41 and equation 32.
13) Page 26, what does $d^\prime$ represent/hold?
14) In most of the experiments, the random sample does almost as good as the given coreset, and in some cases, even better. Do you have an explanation of why this would happen?
15) I suggest the authors use problem-dependent coresets to see how the given coreset is better than a coreset that would aim to approximate the training cost function.
16) I understand that this is the first coreset, and the running time is around $\Omega(n^3)$ which is quite high, however, I suggest the authors provide a graph of time of construction + model fitting for every coreset size for each experiment in the paper.
17) Is there a way to incorporate weights in your approach, i.e., having a weighted coreset?

-----------------------------------
[5] Killamsetty, K., Sivasubramanian, D., Ramakrishnan, G. and Iyer, R., 2021, May. Glister: Generalization based data subset selection for efficient and robust learning. In Proceedings of the AAAI Conference on Artificial Intelligence (Vol. 35, No. 9, pp. 8110-8118).

[6] Tukan, M., Zhou, S., Maalouf, A., Rus, D., Braverman, V. and Feldman, D., 2023, July. Provable data subset selection for efficient neural networks training. In International Conference on Machine Learning (pp. 34533-34555). PMLR.

**Strengths And Weaknesses:**

In what follows, the strengths of the paper are given:
* The first coreset that handles distribution shift, via suggesting DRCS (Distributionally Robust Coreset Selection).
* The idea leverages the test error into the coreset generation process which is a naturally greedy algorithm that would remove the least impactful point iteratively until the number of remaining points is the desired coreset size.

In what follows, the weaknesses of the paper are given:
* The paper is somewhat hard to follow as some terms are ill-defined (see the following section).
* The coreset itself lacks any theoretical guarantees concerning the training data (unlike $\varepsilon$-coresets which are problem-dependant and generated to approximate (multiplicatively) the training cost function of the entire data).
* Considering the papers mentioned in the paper, it is indeed true that they are mostly heuristic, however, for the problems chosen to experiment with, the authors did not choose to test against $\varepsilon$-coresets papers where the coreset is usually a weighted subset. E.g., see [1-4].
* The theoretical running time of either of Algorithms 1,2,3 is not given explicitly.
* The experiments are somewhat weak. Specifically:
  1) In most cases, especially when the coreset size is small (< 20% of the data), uniform sampling performs better than the presented coreset. While it is evident from papers in the fields of $\varepsilon$-coresets, that uniform sampling and the coreset would behave almost the same for large coreset sizes, however, in the small regime after some very small coresets sizes, the proposed coreset should be much better than uniform sampling, showcasing the efficacy and need for alter from random sampling.
2) The experiments were carried out with very small sizes. This defeats the purpose of using a coreset. All data sizes were too small. As for CIFAR10, the data was then post-processed to contain at max 5000 points, which again is too small. One can split the data into mainly three groups where one would be vehicles, the second would be animals, and the remaining can be discarded for example, and run your experiment then on this. The whole idea of a coreset is to handle large-scale datasets, to provide faster training and/or higher quality trained model (due to the data being of higher quality, i.e., lacking noise and incorrectly labeled points for example).

----------------------------------------
[1] Munteanu, A., Schwiegelshohn, C., Sohler, C., & Woodruff, D. (2018). On coresets for logistic regression. Advances in Neural Information Processing Systems, 31.

[2] Tukan, M., Maalouf, A., & Feldman, D. (2020). Coresets for near-convex functions. Advances in Neural Information Processing Systems, 33, 997-1009.

[3] Tukan, M., Baykal, C., Feldman, D. and Rus, D., 2021. On coresets for support vector machines. Theoretical Computer Science, 890, pp.171-191.

[4] Tolochinksy, Elad, Ibrahim Jubran, and Dan Feldman. "Generic coreset for scalable learning of monotonic kernels: Logistic regression, sigmoid and more." International Conference on Machine Learning. PMLR, 2022.

---

> ### Author Response · Authors · 2025-04-09
> **Revisions & Responses on Comments by Reviewer g4Vy (1/2)**
>
> > The paper is somewhat hard to follow as some terms are ill-defined (see the following section).
>
> Thank you for pointing out the parts that makes difficult to follow. We will present improvements in later responses.
>
> > The coreset itself lacks any theoretical guarantees concerning the training data (unlike epsilon-coresets which are problem-dependant and generated to approximate (multiplicatively) the training cost function of the entire data).
>
> The objective of our coreset selection is not to approximate the distribution of the training data, but to identify a subset of the training dataset that is robust to shifts in the test distribution in the sense that the coreset is selected to minimize an upper bound on the test error for this purpose. Therefore, the reviewer's criticism that it lacks theoretical guarantees is a misunderstanding.
>
> > Considering the papers mentioned in the paper, it is indeed true that they are mostly heuristic, however, for the problems chosen to experiment with, the authors did not choose to test against epsilon-coresets papers where the coreset is usually a weighted subset. E.g., see [1-4].
>
> Thank you for providing valuable related researches, and their stategies. In Section 4.1 we added a type of related works "sensitivity-based methods" (to which methods [1-4] belongs), and stated their advantages that the quality of the model parameters are guaranteed as epsilon-coresets. Also, we agree that assigning weights to the coreset may improve our methods (and other methods). So we discuss the strategy as a future direction in Section 6.
>
> > The theoretical running time of either of Algorithms 1,2,3 is not given explicitly.
>
> Thank you for pointing out; we added explicit costs in Section 3.2.
>
> > The experiments are somewhat weak. Specifically: In most cases, especially when the coreset size is small (< 20% of the data), uniform sampling performs better than the presented coreset. While it is evident from papers in the fields of epsilon-coresets, that uniform sampling and the coreset would behave almost the same for large coreset sizes, however, in the small regime after some very small coresets sizes, the proposed coreset should be much better than uniform sampling, showcasing the efficacy and need for alter from random sampling.
>
> We conjecture that the random selection works well than existing methods in a part of settings because the existing methods cannot consider the distributional robustness. We added the discussion about this in Section 5.2.
>
> > The experiments were carried out with very small sizes. This defeats the purpose of using a coreset. All data sizes were too small. As for CIFAR10, the data was then post-processed to contain at max 5000 points, which again is too small. One can split the data into mainly three groups where one would be vehicles, the second would be animals, and the remaining can be discarded for example, and run your experiment then on this. The whole idea of a coreset is to handle large-scale datasets, to provide faster training and/or higher quality trained model (due to the data being of higher quality, i.e., lacking noise and incorrectly labeled points for example).
>
> Unfortunately, our current framework incurs increasing computational costs as the dataset size grows, making it difficult to conduct experiments at the scale suggested by the reviewer at this time. The goal of our distributionally robust coreset selection is to facilitate efficient fine-tuning on edge devices in scenarios where the test distribution varies across individual users. We believe that in coreset selection intended for use on edge devices, methods that are effective on moderately sized datasets are also of significant value.

---

> > ### Author Response · Authors · 2025-04-09
> > **Revisions & Responses on Comments by Reviewer g4Vy (2/2)**
> >
> > > Requested Changes:
> >
> > We fixed the manuscript as follows. Points not described below are fixed as the reviewer indicated.
> >
> > - 2, 3: "\*" denotes the *convex conjugate*, but its definition is lost in the current manuscript. We added it in Section 3.1.1.
> > - 4: "assumption 1" intends the point in the list just above. We rephrased it as "point 1 above".
> > - 5: We intended distribution changes in the class balance, which we often encounter in data analysis; this is why we consider the case when the instance weights are changed from 1 to $a$ for positive instances while unchanged for negative instances. We clarified it in Section 5.1.
> > - 6: Yes, $a = 1 \Rightarrow S = 0$. Here, $S = 0$ means that the distributional robustness is not employed (simple coreset selection problem). We clarified it in Section 5.1.
> > - 7: We intended that we train the feature extractor for the whole data, but for the coreset we train only the last linear layer of the classification (the feature extractor is not updated). This is the limitation of the proposed method about the class of the applicable learning method (Section 2.2). In the revised manuscript, we referred Section 2.2 for the explanation.
> > - 9: It is correct with $f^{\*\*}$, but we explicitly wrote as "$f^{\*\*} (=(f^\*)^\*)$".
> > - 10: Yes, $\otimes$ is the element-wise product. In the revised manuscript we give the definition in Section 3.1.1 (at which it first appears).
> > - 11: "equation 16" (the dual problem without specifying the regularization function) is correct.
> > - 12: The expression itself is correct, but since it is equivalent to equation 41, we unified the notations. Also, in relation to equation 32, we replaced $n_\mathrm{all}^{(\boldsymbol w^\prime)}$ with $n^\prime$ since they are equivalent.
> > - 13: This is a typo: $d^\prime$ should be $d$.
> > - 14: Please see the point "In most cases, especially when the coreset size is small ..." above.
> > - 15: Does it mean that there are datasets whose problem-dependent coresets are easy to obtain?
> > - 16: We understand that $O(n^3)$ cost is quite high, and considering reducing the cost by either of the way: Approximating the $O(n^3)$-time computation, or changing the form of ${\cal W}$ (equation 5) from L2-norm constraint to another for faster computations.
> > - 17: It would be better if our method provides better coresets by not only selecting the instances but also setting weights on the selected instances. Currently we define the selection variable as $\boldsymbol v\in\\{0, 1\\}^n$ in equation (8) (Section 2.3), but replacing it with $\boldsymbol v\in[0, 1]^n$ and applying a sparse optimization on $\boldsymbol v$, it can be regarded as not only the instance selection but also their weights.

---

> > > ### Comment · Reviewer_g4Vy · 2025-04-13
> > > **Thanks for the clarification and for addressing my comments.**
> > >
> > > I thank the authors for the clarification and for addressing my comments.
> > > Concerning question 15: I meant here, using sensitivity-based coresets, which by nature, are problem-dependent (e.g., tailored for logistic regression or SVM).
> > >
> > > As for the scalability-related issues your coreset faces, I find it limiting. However, considering that this is among the first coresets to handle distributional shift, and in fact, it incorporates this problem formulation within the sampling procedure of the coreset, is innovative.
> > >
> > > Another line of coresets that aim to handle similar issues is coresets for continual learning. Assuming that your approach's running time can be better (quadratic time at max via approximation or problem reformulation), how effective would it be against [1] from a practical point of view (quality, not time)?
> > >
> > > ---------------------------------------------
> > >     [1] Borsos, Z., Mutny, M. and Krause, A., 2020. Coresets via bilevel optimization for continual learning and streaming. Advances in neural information processing systems, 33, pp.14879-14890.

---

> > > > ### Author Response · Authors · 2025-04-16
> > > >
> > > > First, we understood the intention of question 15. It is a good way to examine the method in more detail, although we may need to prepare such datasets by considering not only the training method dependency but also the distributional robustness.
> > > >
> > > > Also, thank you for providing interesting method (Borsos 2020). Although we have not fully understood, we felt it interesting since it can provide a theoretical assurance of the *optimality gap* of the objective function after the coreset selection (Theorem 1 in the paper).
> > > > Since our method provides assurances of the model parameters after the coreset selection by first computing the *duality gap* of the objective function and then converting it to the maximum change of the model parameters, we may replace the duality gap with the one in (Borsos 2020).
> > > > Another interesting point of (Borsos 2020) is that, the algorithm is formulated so that a greedy algorithm to select the coreset can provide a theoretical assurance, thanks to (Locatello 2017).
> > > > So, extending it to distributionally robust setting may be an interesting direction, although we need to examine how the cost is increased to handle distributionally robust setting.
> > > >
> > > > - Note: the "optimality gap" is the difference of the objective function from the true optimum, while the "duality gap" is the difference of the objective function between the primal and the dual problem. The latter is larger.
> > > >
> > > > (Locatello 2017) F. Locatello, M. Tschannen, G. Rätsch, and M. Jaggi. Greedy algorithms for cone constrained optimization with convergence guarantees. In Advances in Neural Information Processing Systems, pages 773–784, 2017.

---

### Review · Reviewer_Mf39 · 2025-03-16

**Summary Of Contributions:**

Coreset selection is an approach used in machine learning to select a small representative subset of a training dataset while preserving model performance. In coreset selection, it is generally assumed that the training and test data distributions are the same i.e., they follow the same underlying probability distribution. The paper ``Distributionally Robust Coreset Selection under Covariate Shift'' introduces a novel method (called DRCS) to address the challenge of selecting a coreset of training data under a scenario where the input distribution at deployment differs from training. The key contribution of DRCS is a theoretical upper bound for the worst-case test error, which guides the selection of a coreset that maintains model performance under uncertain distribution shifts. In a conventional covariate shift setting, the test input distribution is assumed to be known, allowing weights $\omega$ to be determined based on the density ratio between test and training distributions. In the distributionally robust setting, the density ratio (and thus $ \omega $) is unknown but is assumed to lie within a defined uncertainty set:
$$W :=  \{ w \in \mathbb{R}^n \mid \| w - \mathbf{1}_n \|_2 \leq S \}$$
where $S$ controls the degree of allowed deviation from uniform weights. The final DRCS optimization problem is designed as a trilevel optimization problem with the third level as a combinatorial optimization which becomes infeasible for large cases.
To make the selection process computationally efficient, the authors propose (1) an upper bound for the worst-case weighted validation error and (2)  three greedy heuristics that approximate the optimal coreset selection. Through experiments on tabular and image datasets, DRCS demonstrates superior performance compared to existing coreset selection methods, proving its robustness and practicality.

**Audience:**

Yes

**Broader Impact Concerns:**

No Broader Impact Concerns.

**Claims And Evidence:**

Yes

**Requested Changes:**

It would be helpful for the authors to either address the weaknesses outlined above under "Weaknesses" or provide an explanation if addressing them is particularly challenging.

**Strengths And Weaknesses:**

Strengths:

The paper presents a novel and rigorous approach to coreset selection under covariate shift, addressing a challenge in distributionally robust learning. The paper derives an upper bound for the worst-case test error, providing a rigorous guarantee on the robustness of coreset selection under uncertain test distributions. It introduces a trilevel optimization framework to formally model the worst-case risk minimization problem, accounting for both coreset selection and distributional shifts. The use of duality theory and strong convexity assumptions enables an efficient computation of the worst-case bound.  The paper identifies the computational intractability of the exact trilevel optimization problem and proposes efficient greedy heuristics to approximate optimal coreset selection. To efficiently compute the worst-case test error bound, the upper bound of the weighted validation error is maximized with respect to the weight vector $\omega’$ for the validation dataset. This maximization is performed through eigenvalue solving under $L_2$ regularization, allowing to derive an upper bound for the worst-case weighted validation error.  The authors conduct extensive experiments on tabular and image datasets, demonstrating that DRCS outperforms existing coreset selection methods in terms of robustness to distributional shifts.

Weaknesses:
1. One of the primary limitations of the paper is the absence of a formal theoretical analysis quantifying how tight or loose the proposed upper bound on the worst-case test error is. While the bound ensures robustness, it remains unclear how closely it approximates the true worst-case error in practical settings. An empirical study comparing the upper bound with actual worst-case errors would further validate the robustness and practical applicability of the proposed methods. This could involve evaluating the computed worst-case validation error bound alongside the empirically observed worst-case validation error, providing insight into how well the bound reflects real performance.

2. A recent paper examined coresets for classification in a regularized setting:

Meysam Alishahi and Jeff M. Phillips. 2024. "No Dimensional Sampling Coresets for Classification." In Proceedings of the 41st International Conference on Machine Learning (ICML'24), Vol. 235. JMLR.org, Article 43, pp. 1008–1049.

Given the similarity in setting, the authors may find it useful to refer to this paper and explore potential connections between its results and their own findings. Furthermore, a promising direction for future study would be to investigate whether it is possible to establish a dimension-free bound for the worst-case test error.

---

> ### Author Response · Authors · 2025-04-09
> **Revisions & Responses on Comments by Reviewer Mf39**
>
> > One of the primary limitations of the paper is the absence of a formal theoretical analysis quantifying how tight or loose the proposed upper bound on the worst-case test error is. While the bound ensures robustness, it remains unclear how closely it approximates the true worst-case error in practical settings. An empirical study comparing the upper bound with actual worst-case errors would further validate the robustness and practical applicability of the proposed methods. This could involve evaluating the computed worst-case validation error bound alongside the empirically observed worst-case validation error, providing insight into how well the bound reflects real performance.
>
> First, we did show the comparison of the prediction performances between theoretical and experimental results in Section 5 (lines "Guarantee" in Figures 4 and 7). This showed that the guaranteed performances were much smaller than the true ones.
>
> This is because the proposed bounds are not probabilistic but deterministic, and therefore difficult to produce tight bounds. In order to provide a tighter bound, applying a probabilistic analysis may be helpful, and we stated it as a future direction in Section 6 in the revised manuscript.
>
> > A recent paper examined coresets for classification in a regularized setting:
> > Meysam Alishahi and Jeff M. Phillips. 2024. "No Dimensional Sampling Coresets for Classification." In Proceedings of the 41st International Conference on Machine Learning (ICML'24), Vol. 235. JMLR.org, Article 43, pp. 1008–1049.
> > Given the similarity in setting, the authors may find it useful to refer to this paper and explore potential connections between its results and their own findings. Furthermore, a promising direction for future study would be to investigate whether it is possible to establish a dimension-free bound for the worst-case test error.
>
> This paper is interesting both in the procedure and the analysis. Especially, the analysis procedure that mainly based on probability analysis is quite different from ours. It will be better if such a probability analysis is applied to our methods. We cited the paper to state as a future direction of our work in Section 6 in the revised manuscript.

---

> > ### Comment · Reviewer_Mf39 · 2025-05-09
> > **Thanks for the clarification.**
> >
> > Thank you for your clarification on the comparison between theoretical and experimental prediction performance. I'm also pleased to see the suggested reference cited and acknowledged as a future direction. I’m satisfied with your revisions and responses.

---

### Review · Reviewer_EyTh · 2025-03-26

**Summary Of Contributions:**

This paper proposes Distributionally Robust Coreset Selection (DRCS) to address the problem of selecting a small subset of training data (a coreset) when there is a distribution shift between the deployment environment and the training one (i.e., covariate shift). As the data distribution during deployment is often unknown, selecting an effective subset of training data that performs reasonably well across all possible test distributions is a challenge. The paper derives an upper bound for the worst-case test error and designs DRCS that suppresses the estimate of this upper bound for the worst-case test error. The experiment is run on both tabular and image data and shows the effectiveness of DRCS.

**Audience:**

Yes

**Broader Impact Concerns:**

Not applicable.

**Claims And Evidence:**

Yes

**Requested Changes:**

Could the authors discuss the connection between coreset selection and dataset distillation?

Could the authors explain how to extend the work to non-convex functions?

Please also give the computational cost of the proposed method along with baselines'.

**Strengths And Weaknesses:**

Strengths: The paper proposes to address a key problem in distribution shift robustness. The theoretical result looks rigorous. Fig. 1-2 are helpful for understanding.

Weaknesses: The strong convexity assumption is an obvious weakness. The applicable task is only binary classification. The above two weaknesses limit the scope of this paper. Calculating the upper bound of the worst-case weighted validation error can be computationally expensive.

---

> ### Author Response · Authors · 2025-04-09
> **Revisions & Responses on Comments by Reviewer EyTh**
>
> > Weaknesses: The strong convexity assumption is an obvious weakness. The applicable task is only binary classification. The above two weaknesses limit the scope of this paper. Calculating the upper bound of the worst-case weighted validation error can be computationally expensive.
>
> First, for the limitation of strong convexity, please refer the response for "Could the authors explain how to extend the work to non-convex functions?" below.
>
> Secondly, the limitation to the classification problems comes from the need to compute the upper bound of the validation error. However, in reality, we may also compute the upper bound of the validation error in regression problems. In order to compute the upper bound of the validation error, we just used the upper bound of the linear prediction value ("$\\ldots$" of $I\\{\\ldots > 0\\}$ in equation 19), which can be also used for regression problems. We added in Section 6 in the revised manuscript.
>
> Lastly, we understand that $O(n^3)$ cost to compute the worst-case weighted validation error is quite high, and considering reducing the cost by either of the way: Approximating the $O(n^3)$-time computation, or changing the form of ${\cal W}$ (equation 5) from L2-norm constraint to another for faster computations. We added a discussion in Section 3.2 in the revised manuscript.
>
> > Could the authors discuss the connection between coreset selection and dataset distillation?
>
> We understand the relationship to dataset distillation (that usually generates new instances rather than just selecting from existing instances), although the extension from our method may not be straightforward in the sense that the optimization procedure is assumed to be much more complex. We stated in Section 6 in the revised manuscript.
>
> > Could the authors explain how to extend the work to non-convex functions?
>
> First, we limit the proposed method to the strongly convex learning methods since it is needed to assure the upper bound of the change in the model parameter. If we apply our method to convex (but not strongly convex) or non-convex learning methods, we need to solve the following issues.
>
> - If we may omit the assurance in the validation error,
>   - If the learning method is convex, then we have only to apply the proposed method, since our coreset selection algorithm itself (Appendix C.4) assumes the existence of the dual problem but not of the strong convexity.
>   - If the learning method is non-convex, we also need to derive an alternative of the dual problem; in other words, a function that works as a lower bound of the objective function.
> - If we need an assurance in the validation error, a possible direction is to divide the objective function into strongly-convex pieces.
>
> > Please also give the computational cost of the proposed method along with baselines'.
>
> We did not show them since it cannot be a fair comparison; the baseline methods do not consider distributional robustness and therefore works faster than the proposed method.

---

> > ### Comment · Reviewer_EyTh · 2025-05-09
> >
> > Thanks for the response. I still think the computational cost and the strong convexity assumption are weaknesses of the paper. However, the evidence to support the existing claims is high quality, so I won't recommend rejection.

---

### Author Response · Authors · 2025-04-09
**Responses to all reviewers**

We would like to thank three reviewers for fruitful comments.
We uploaded a revised manuscript in which all fixes corresponding to reviewers' comments are written in red text.

---

### Decision · Action_Editor_iM6k · 2025-05-24

**Recommendation:** Accept as is

**Comment:**

The paper is on topic for the audience, formulates an interesting problem space, and provides useful results in addressing the problem.  I reviewers agree it fits the requirements for TMLR after the responses.  There are no outstanding issues, so can be accepted as is.

**Audience:**

This paper is clearly within the scope of TMLR, and reviewers agree.

**Claims And Evidence:**

All reviewers agree that the claims are supported by sufficient evidence.  I concur.